# Fair Representation in Submodular Subset Selection: A Pareto Optimization Approach

**Adriano Fazzone**                                              *adriano.fazzone@centai.eu*
*CENTAI Institute, Turin, Italy*

**Yanhao Wang**                                                 *yhwang@dase.ecnu.edu.cn*
*East China Normal University, Shanghai, China*

**Francesco Bonchi**                                            *francesco.bonchi@centai.eu*
*CENTAI Institute, Turin, Italy*
*Eurecat, Barcelona, Spain*

**Reviewed on OpenReview:** *https://openreview.net/forum?id=0Hm01Vc8zT*

## Abstract

Many machine learning applications, such as feature selection, recommendation, and social advertising, require the joint optimization of the global *utility* and the *representativeness* for different groups of items or users. To meet such requirements, we propose a novel multi-objective combinatorial optimization problem called *Submodular Maximization with Fair Representation* (SMFR), which selects subsets from a ground set, subject to a knapsack or matroid constraint, to maximize a submodular (*utility*) function $f$ as well as a set of $d$ submodular (*representativeness*) functions $g_1, \ldots, g_d$. We show that the maximization of $f$ might conflict with the maximization of $g_1, \ldots, g_d$, so that no single solution can optimize all these objectives at the same time. Therefore, we propose a Pareto optimization approach to SMFR, which finds a set of solutions to approximate all Pareto-optimal solutions with different trade-offs between the objectives. Our method converts an instance of SMFR into several submodular cover instances by adjusting the weights of the objective functions; then it computes a set of solutions by running the greedy algorithm on each submodular cover instance. We prove that our method provides approximation guarantees for SMFR under knapsack or matroid constraints. Finally, we demonstrate the effectiveness of SMFR and our proposed approach in two real-world problems: *maximum coverage* and *recommendation*.

## 1 Introduction

The problem of subset selection aims to pick a maximum utility subset $S$, under a given constraint, from a ground set $V$ of items. This fundamental problem arises in a wide range of machine learning applications, such as social advertising (Kempe et al., 2003; Aslay et al., 2015; 2017; Tang, 2018), recommendation systems (Ohsaka & Matsuoka, 2021; Mehrotra & Vishnoi, 2023), data summarization (Lin & Bilmes, 2010; Mirzasoleiman et al., 2016), and feature selection (Liu et al., 2013; Bao et al., 2022), to name just a few. A common combinatorial structure in such problems is *submodularity* (Krause & Golovin, 2014), which naturally captures the "diminishing returns" property: adding an item to a smaller set produces a higher marginal gain than adding it to a larger set. This property not only captures the desirable properties of *coverage* and *diversity* of subsets, but also enables the design of efficient approximation algorithms.

Among the various combinatorial optimization problems for subset selection in the literature, maximizing a monotone submodular function subject to a knapsack constraint (SMK) or a matroid constraint (SMM) has attracted a lot of attention, as such constraints capture common scenarios in which the selected subset must be limited within a budget (Krause & Guestrin, 2005; Călinescu et al., 2011).

More formally, given a ground set $V$ of $n$ items, we consider a set function $f : 2^V \to \mathbb{R}^+$ to measure the *utility* $f(S)$ of any set $S \subseteq V$. We assume that $f$ is normalized, i.e., $f(\emptyset) = 0$, monotone, i.e., $f(S) \leq f(T)$ for any $S \subseteq T \subseteq V$, and submodular, $f(S \cup \{v\}) - f(S) \geq f(T \cup \{v\}) - f(T)$ for any $S \subseteq T \subseteq V$ and $v \in V \setminus T$. We also consider a cost function $c : V \to \mathbb{R}^+$ which assigns a positive cost $c(v)$ to each item $v \in V$, and we denote $c(S)$ the cost of a set $S \subseteq V$, defined as the sum of costs for all items in $S$, i.e., $c(S) = \sum_{v \in S} c(v)$. For a given budget $k \in \mathbb{R}^+$, the set of all feasible solutions subject to the knapsack constraint contains all subsets of $V$ whose costs are at most $k$, i.e., $\mathcal{I}_k = \{S \subseteq V : c(S) \leq k\}$. The SMK problem on $f$ is defined as $S^*_f = \arg\max_{S \in \mathcal{I}_k} f(S)$. Furthermore, a matroid $\mathcal{M}$ on a ground set $V$ is defined by a collection $\mathcal{I}(\mathcal{M})$ of subsets of $V$ called the *independent sets*, that satisfies the following properties: (1) $\emptyset \in \mathcal{I}(\mathcal{M})$; (2) for any $S \subset T \subseteq V$, if $T \in \mathcal{I}$, then $S \in \mathcal{I}(\mathcal{M})$ holds; (3) for any $S, T \subseteq V$, if $|S| < |T|$, there exists $v \in T \setminus S$ such that $S \cup \{v\} \in \mathcal{I}(\mathcal{M})$. Here, the size of the maximum independent sets in $\mathcal{M}$ is called its rank $r(\mathcal{M})$. Similarly to SMK, the SMM problem on $f$ is defined as $S^*_f = \arg\max_{S \in \mathcal{I}(\mathcal{M})} f(S)$.

In many real-world problems, in addition to the primary objective of maximizing the utility function $f$, it is often essential to take into account the representativeness with respect to different groups of items or users. For example, consider the influence maximization problem (Tsang et al., 2019; Becker et al., 2020):

**Example 1.** Let $\mathcal{G} = (V, E)$ be a graph that denotes the relationships between a set of users $V$ on a social network. Each user $v \in V$ is also associated with a sensitive attribute $\mathcal{A}$ to divide $V$ into multiple protected groups. The influence maximization (IM) problem (Kempe et al., 2003) aims to select a subset $S \subseteq V$ of users as seeds to maximize a (monotone, submodular) influence spread function under an information diffusion (e.g., independent cascade or linear threshold) model. If the information to be spread is related to education and employment opportunities, fair access to information between protected groups (Tsang et al., 2019; Becker et al., 2020) becomes a critical issue. This is often formulated as maximizing the influence spread functions specific to all protected groups in a balanced manner so that none of the groups is much worse off than the others. Furthermore, we should also impose constraints in different contexts on the seed set $S$, e.g., to limit the overall budget for the propagation campaign, the total cost of $S$ should be within an upper bound (*knapsack constraint*), or to ensure diversity, the number of seeds in $S$ from any demographic category cannot exceed an upper limit (*matroid constraint*).

The above problem, as well as many other subset selection problems with fairness or other representativeness considerations (Krause et al., 2008; Mirzasoleiman et al., 2016; Wang et al., 2024), can be formulated as a multi-objective optimization problem of selecting a set $S$ to simultaneously maximize a monotone submodular *utility* function $f$ and a set of $d$ monotone submodular *representativeness* functions $g_1, \ldots, g_d$, all defined on the same ground set $V$, subject to a knapsack or matroid constraint $\mathcal{I}$:

$$\max_{S \in \mathcal{I}} (f(S), g_1(S), \ldots, g_d(S)). \tag{1}$$

We call this problem *Submodular Maximization with Fair Representation* (SMFR)[1] since it captures the case where the submodular utility function $f$ and all the submodular representativeness functions $g_1, \ldots, g_d$ are maximized at the same time to avoid under-representing any of them.

**Our Contributions.** To the best of our knowledge, SMFR is a novel optimization problem, never addressed before (see Section 2 for a detailed discussion of how the related literature differs from SMFR). It is easy to see that SMFR is at least as hard as SMK and SMM, which cannot be approximated within a factor better than $1 - 1/e$ unless $P = NP$ (Feige, 1998). However, SMFR is much more challenging than SMK and SMM due to its multi-objective nature. By providing a counterexample (see Example 2), we show that there might not exist any single solution to an instance of SMFR that achieves an approximation factor greater than 0 to maximize $f$ and $g_1, \ldots, g_d$ at the same time, even when $d = 1$. Due to the inapproximability of SMFR, we consider approaching it by *Pareto optimization*. Specifically, we call a set $S$ an $(\alpha, \beta)$-approximate solution for an instance of SMFR if $S \in \mathcal{I}$, $f(S) \geq \alpha \texttt{OPT}_f$, where $\texttt{OPT}_f = \max_{S' \in \mathcal{I}} f(S')$, and $g_i(S) \geq \beta \texttt{OPT}_{g_i}$ for all $i = 1, \ldots, d$, where $\texttt{OPT}_{g_i} = \max_{S' \in \mathcal{I}} g_i(S')$. An $(\alpha, \beta)$-approximate solution $S$ is Pareto optimal if there does not exist an $(\alpha', \beta')$-approximate solution $S' \in \mathcal{I}$ for any $\alpha' \geq \alpha, \beta' \geq \beta$ and at least one is strictly larger. Since computing a single Pareto optimal solution to SMFR is still NP-hard, we turn our attention to identifying a set $\mathcal{S}$ of multiple solutions to approximate the *Pareto frontier*; that is, to find

---

[1] Note that all objective functions are unordered and equally important in Eq. 1.

a set $\mathcal{S}$ such that for any Pareto-optimal solution, there exists a corresponding solution in $\mathcal{S}$ achieving a bounded approximation for it. Our framework first uses any existing algorithm for SMK (Sviridenko, 2004; Yaroslavtsev et al., 2020; Tang et al., 2021; Feldman et al., 2022; Li et al., 2022) or SMM (Fisher et al., 1978; Vondrak, 2008; Călinescu et al., 2011; Badanidiyuru & Vondrák, 2013; Filmus & Ward, 2014; Buchbinder et al., 2019) to approximate $\mathtt{OPT}_f$ and each $\mathtt{OPT}_{g_i}$. Based on the approximations, our proposal transforms an instance of SMFR into multiple instances of the submodular cover problem with different weights on $\mathtt{OPT}_f$ and each $\mathtt{OPT}_{g_i}$ to capture the trade-offs between $f$ and each $g_i$. Then, classic greedy algorithms (Wolsey, 1982; Torrico et al., 2021) are used to obtain an approximate solution for each submodular cover instance. Finally, all the above-computed solutions that are not "dominated"[2] by any other computed solution are returned as the set $\mathcal{S}$ of at most $O(\frac{1}{\varepsilon})$ approximate solutions to SMFR for any $\varepsilon \in (0, 1)$. Theoretically, our framework provides approximation bounds for SMFR under both knapsack and matroid constraints:

- When using a $\delta$-approximation algorithm for SMK, it provides a set $\mathcal{S}$ such that for each $(\alpha, \beta)$-approximate Pareto optimal solution of SMFR, there must exist a corresponding $(\delta\alpha - \varepsilon, \delta\beta - \varepsilon)$-approximate solution of cost $O(k \log \frac{d}{\varepsilon})$ in $\mathcal{S}$, where $k \in \mathbb{R}^+$ is the budget of the knapsack constraint.
- When using a $\delta$-approximation algorithm for SMM, it also provides a set $\mathcal{S}$ such that for each $(\alpha, \beta)$-approximate Pareto optimal solution of SMFR, there must exist a corresponding $(\delta\alpha - \varepsilon, \delta\beta - \varepsilon)$-approximate solution of size $O(r \log \frac{d}{\varepsilon})$ in $\mathcal{S}$, where $r \in \mathbb{Z}^+$ is the rank of the matroid constraint.

In the empirical assessment, we evaluate our proposed framework for the problems of *maximum coverage* and *recommendation* using real-world data. The numerical results confirm the effectiveness of our proposal compared to competitive baselines.

**Paper Organization.** The remainder of this paper is organized as follows. We review the related work in Section 2. Then, we analyze the hardness of SMFR in Section 3. Next, our algorithmic framework for SMFR is presented in Section 4. Subsequently, the experimental setup and results are provided in Section 5. Finally, we conclude the paper and discuss future work in Section 6. The proofs of theorems and lemmas and several supplemental experiments are deferred to the appendices due to space limitations.

## 2 Related Work

**Monotone Submodular Maximization with Knapsack or Matroid Constraints.** There exists a wide literature on maximizing a monotone submodular function subject to a knapsack constraint (SMK) or a matroid constraint (SMM). For cardinality constraints, a special case of both knapsack and matroid constraints, Nemhauser et al. (1978) proposed a simple greedy algorithm that runs in $O(kn)$ time and yields the best possible approximation factor $1 - 1/e$ unless $P = NP$. However, the greedy algorithm can be arbitrarily bad for general knapsack or matroid constraints. Sviridenko (2004) first proposed a greedy algorithm with partial enumerations that achieves the best possible approximation $1 - 1/e$ for SMK in $O(n^5)$ time. Kulik et al. (2021) and Feldman et al. (2022) improved the time complexity to $O(n^4)$ while keeping the same approximation factor. Krause & Guestrin (2005) proposed an $O(n^2)$-time $\frac{1}{2}(1 - \frac{1}{e}) \approx 0.316$-approximation cost-effective greedy algorithm for SMK. Tang et al. (2021), Kulik et al. (2021), and Feldman et al. (2022) improved the approximation factor of the cost-effective greedy algorithm to 0.405, $[0.427, 0.4295]$, and $[0.427, 0.462]$ independently. Ene & Nguyen (2019a) proposed a near-linear time $(1 - 1/e - \varepsilon)$-approximation algorithm for SMK based on multilinear relaxation. Yaroslavtsev et al. (2020) proposed a $\frac{1}{2}$-approximation Greedy+Max algorithm for SMK in $O(n^2)$ time. Feldman et al. (2022) further provided an approximation factor of 0.6174 in $O(n^3)$ time by enumerating each item as a partial solution and running Greedy+Max on each partial solution. Li et al. (2022) recently proposed a $(\frac{1}{2} - \varepsilon)$-approximation algorithm for SMK in $O(\frac{n}{\varepsilon} \log \frac{1}{\varepsilon})$ time. Fisher et al. (1978) first proposed a $\frac{1}{2}$-approximation greedy algorithm for SMM running in $\tilde{O}(nr)$ time. Călinescu et al. (2011) and Vondrak (2008) independently proposed randomized continuous greedy algorithms with rounding for SMM. Both algorithms achieved the best possible $(1 - 1/e)$-approximation in expectation but had prohibitive $O(n^8)$ running time. Badanidiyuru & Vondrák (2013) proposed a faster continuous greedy algorithm that yielded a $(1 - 1/e - \varepsilon)$-approximation for SMM in

---

[2]A solution $S$ will be dominated by another solution $T$ if the approximation factors $\alpha, \beta$ of $S$ are both no greater than those of $T$ and at least one is strictly smaller.

$O(\frac{n^2}{\varepsilon^4} \log^2 \frac{n}{\varepsilon})$ time. Filmus & Ward (2014) proposed a $(1 - 1/e - \varepsilon)$-approximation algorithm in $O(\frac{nr^4}{\varepsilon^3})$ time and a $(1 - 1/e)$-approximation algorithm in $O(n^2 r^7)$ time, both randomized and based on non-oblivious local search. Buchbinder et al. (2019) proposed the first deterministic algorithm for SMM with an approximation factor over $1/2$ in $O(nr^2)$ time. Ene & Nguyen (2019b) also proposed a nearly linear-time $(1 - 1/e - \varepsilon)$-approximation algorithm for SMM based on multilinear relaxation. Although the above algorithms cannot be applied directly to SMFR, any of them can serve as a subroutine in our algorithmic framework for SMFR.

**Multi-objective Submodular Maximization.** There exist several variants of submodular maximization problems to deal with more than one objective. Next, we discuss multi-objective submodular maximization problems relevant to SMFR. One basic problem of this kind is to maximize a weighted sum of $d > 1$ submodular functions $g_1, \ldots, g_d$. Since the weighted sum of multiple submodular functions is still submodular (Krause & Golovin, 2014), this problem can be directly resolved with any algorithm for submodular maximization. However, maximizing the weighted sum is often not enough to achieve a fair representation among all objectives, as its (optimal) solution may not have any approximation guarantee for each objective function individually. The problem of maximizing the minimum of $d > 1$ submodular functions $g_1, \ldots, g_d$ was studied in (Krause et al., 2008; Udwani, 2018; Anari et al., 2019; Torrico et al., 2021). This problem differs from SMFR because it does not consider maximizing $f$ and aims to return only a single solution for all functions. Nevertheless, we draw inspiration from the SATURATE framework first proposed by Krause et al. (2008) to address SMFR. Another two relevant problems to SMFR are *Submodular Maximization under Submodular Cover* (SMSC) (Ohsaka & Matsuoka, 2021), which maximizes one submodular function subject to the value of the other submodular function not being below a threshold, and *Balancing utility and fairness in Submodular Maximization* (BSM) (Wang et al., 2024), which maximizes a submodular utility function subject to that a fairness function in form of the minimum of $d > 1$ submodular functions is approximately maximized. SMSC and BSM differ from SMFR in the following four aspects: ($i$) they also return a single solution to optimize a user-specified trade-off between multiple objectives; ($ii$) they are specific to cardinality constraints but cannot handle more general knapsack or matroid constraints; ($iii$) SMSC is limited to two submodular functions, that is, a special case of $d = 1$ in SMFR; ($iv$) BSM requires all objective functions to be decomposable. Thus, SMFR can work in more general scenarios than SMSC and BSM. Due to the above differences, the algorithms for SMSC and BSM cannot be used directly for SMFR, and in the experiments they will be compared with our algorithm after adaptations. Very recently, Tang & Yuan (2023) proposed a randomized subset selection method to maximize a (submodular) overall utility function while the (submodular) utility functions for $d$ groups are all not below a lower bound in expectation. They also considered submodular maximization with group equality, which ensures that the difference in the expected utilities of any two groups does not exceed an upper bound. As they limit their consideration to cardinality constraints and their problem formulations are different from SMFR, their proposed methods do not apply to SMFR. The problem of regret-ratio minimization (Soma & Yoshida, 2017; Feng & Qian, 2021; Wang et al., 2023) for multi-objective submodular maximization is similar to SMFR in the sense that they also aim to find a set of approximate solutions for different trade-offs between multiple objectives. However, they consider denoting the trade-offs as different non-negative linear combinations of multiple submodular functions but cannot guarantee any approximation for each objective individually.

Finally, several subset selection problems, e.g., (Qian et al., 2015; 2017; 2020; Roostapour et al., 2022), utilize a Pareto optimization method by transforming a single-objective problem into a bi-objective problem and then solving the bi-objective problem to obtain an approximate solution to the original problem. These problems are interesting but orthogonal to our work.

## 3 Hardness of SMFR

In this paper, we focus on the SMFR problem in Eq. 1 subject to a knapsack or matroid constraint. Next, we formally analyze the computational hardness of SMFR. Since SMK and SMM are both NP-hard and cannot be approximated within a factor $1 - 1/e + \varepsilon$ in polynomial time for any $\varepsilon > 0$ unless $P = NP$ (Feige, 1998; Khuller et al., 1999), the problem of maximizing $f$ or each $g_i$ individually can only be solved approximately. We provide a trivial example to indicate that the maximization of $f$ and the maximization of each $g_i$ could conflict with each other, and there might not exist any $S \in \mathcal{I}$ with approximation factors greater than 0 for both of them, even when $d = 1$.

**Example 2.** Suppose that $d = 1$ and the set of feasible solutions $\mathcal{I}$ is defined by a cardinality constraint 1, i.e., $\mathcal{I} = \{S \subseteq V : |S| \leq 1\}$. Note that a cardinality constraint is a special case of both knapsack and matroid constraints. For the two functions $f$ and $g_1$, we have $\texttt{OPT}_f = f(\{v_0\}) = 1$, $\texttt{OPT}_{g_1} = g_1(\{v_1\}) = 1$, $g_1(\{v_0\}) = 0$, $f(\{v_1\}) = 0$, and $f(\{v_j\}) = g_1(\{v_j\}) = 0$ for any $j > 1$. In the above SMFR instance, there is no set $S \in \mathcal{I}$ such that $f(S) > 0$ and $g_1(S) > 0$.

Given the above result, we are motivated to introduce *Pareto optimization*, a well-known concept for multi-objective optimization (Qian et al., 2015; Soma & Yoshida, 2017) which provides more than one solution with different (best possible) trade-offs between multiple objectives. We call a set $S \in \mathcal{I}$ an $(\alpha, \beta)$-approximate solution for an instance of SMFR if $f(S) \geq \alpha\texttt{OPT}_f$ and $g_i(S) \geq \beta\texttt{OPT}_{g_i}, \forall i \in [d]$. An $(\alpha, \beta)$-approximate solution $S$ is Pareto optimal if there does not exist an $(\alpha', \beta')$-approximate solution for any $\alpha' \geq \alpha$ and $\beta' \geq \beta$ and at least one is strictly larger. Ideally, by enumerating all distinct Pareto optimal solutions (which form the so-called *Pareto frontier*), one can obtain all different optimal trade-offs between maximizing $f$ and each $g_i$. However, computing any Pareto optimal solution is still NP-hard. To circumvent the barrier, a feasible approach to SMFR is to find a set $\mathcal{S}$ of approximate solutions, in which, for any Pareto optimal solution, at least one solution close to it is included. This is the approach that we follow in our framework.

## 4 The SMFR-Saturate Framework

To find approximate solutions to an instance of SMFR, we propose to transform it into a series of instances of its corresponding decision problems, that is, to determine whether there exists any $(\alpha, \beta)$-approximate solution for the SMFR instance. Then, we introduce the SATURATE framework first proposed in (Krause et al., 2008) to approximately solve each instance of the decision problem as *Submodular Cover* (SC), that is, the problem of finding a set $S_c^*$ with the minimum cardinality/cost such that $f(S_c^*) \geq L$ for some $L \in \mathbb{R}^+$. Now, we formally define the decision problem and analyze why the transformation follows.

**Definition 1** (SMFR-DEC). Given an instance of SMFR and two approximation factors $\alpha, \beta \in [0, 1]$, find a set $S \in \mathcal{I}_k$ such that $f(S) \geq \alpha\texttt{OPT}_f$ and $g_i(S) \geq \beta\texttt{OPT}_{g_i}$ for each $i \in [d]$, or decide that there does not exist any set that can meet the conditions.

Assuming that $\texttt{OPT}_f$ and each $\texttt{OPT}_{g_i}$ are already known, the above conditions can be equivalently expressed as $\frac{f(S)}{\alpha\texttt{OPT}_f} \geq 1$ and $\frac{g_i(S)}{\beta\texttt{OPT}_{g_i}} \geq 1$. Then, using the truncation technique in (Krause et al., 2008), SMFR-DEC is converted to decide whether the objective value of the following problem is $d + 1$:

$$\max_{S \in \mathcal{I}} F_{\alpha,\beta}(S) := \min\left\{1, \frac{f(S)}{\alpha\texttt{OPT}_f}\right\} + \sum_{i=1}^{d} \min\left\{1, \frac{g_i(S)}{\beta\texttt{OPT}_{g_i}}\right\}. \tag{2}$$

Note that $F_{\alpha,\beta}$ is ill-formulated due to division by zero when $\alpha$, $\beta$ or $\texttt{OPT}_f$, $\texttt{OPT}_{g_i}$ are equal to 0. To solve this problem, the first term of $F_{\alpha,\beta}$ is replaced by 1 when $\alpha = 0$ or $\texttt{OPT}_f = 0$; the second term of $F_{\alpha,\beta}$ is replaced by $d$ when $\beta = 0$ or $\texttt{OPT}_{g_i} = 0$ for any $i \in [d]$.

The above conversion holds because $F_\alpha(S) = d + 1$ if and only if $f(S) \geq \alpha\texttt{OPT}_f$ and $g_i(S) \geq \beta\texttt{OPT}_{g_i}, \forall i \in [d]$. In addition, $F_{\alpha,\beta}$ is a normalized, monotone, and submodular function because the minimum of a positive real number and a monotone submodular function is monotone and submodular (Krause et al., 2008), and the nonnegative linear combination of monotone submodular functions is monotone and submodular (Krause & Golovin, 2014). In this way, SMFR-DEC is transformed to SC on $F_{\alpha,\beta}$.

Since computing $\texttt{OPT}_f$ and $\texttt{OPT}_{g_i}$ is NP-hard, we should use any existing algorithm for SMK (Sviridenko, 2004; Yaroslavtsev et al., 2020; Tang et al., 2021; Feldman et al., 2022; Li et al., 2022) or SMM (Fisher et al., 1978; Vondrak, 2008; Călinescu et al., 2011; Badanidiyuru & Vondrák, 2013; Filmus & Ward, 2014; Buchbinder et al., 2019) to compute their approximations. Suppose that we run an approximation algorithm for SMK or SMM to obtain $\texttt{OPT}_f' \leq \texttt{OPT}_f$ and $\texttt{OPT}_{g_i}' \leq \texttt{OPT}_{g_i}, \forall i \in [d]$ accordingly. The problem in Eq. 2 is relaxed as follows:

$$\max_{S \in \mathcal{I}} F_{\alpha,\beta}'(S) := \min\left\{1, \frac{f(S)}{\alpha\texttt{OPT}_f'}\right\} + \sum_{i=1}^{d} \min\left\{1, \frac{g_i(S)}{\beta\texttt{OPT}_{g_i}'}\right\}, \tag{3}$$

---

**Algorithm 1:** SMFR-SATURATE

---

**Input:** Normalized, monotone, and submodular set functions $f, g_1, \ldots, g_d : 2^V \to \mathbb{R}^+$; Cost function $c :$ $V \to \mathbb{R}^+$ and budget $k \in \mathbb{R}^+$ (for knapsack constraint) or Collection of feasible sets $\mathcal{I}(\mathcal{M}) \subseteq 2^V$ and rank $r \in \mathbb{Z}^+$ (for matroid constraint); Error parameter $\varepsilon \in (0, 1)$

**Result:** A set $\mathcal{S}$ of approximate solutions to SMFR

Initialize $\mathcal{S} \leftarrow \emptyset$;

Run an algorithm for SMK or SMM to maximize $f, g_1, \ldots, g_d$ subject to the constraint $\mathcal{I}_k$ or $\mathcal{I}(\mathcal{M})$ to compute $\mathtt{OPT}'_f, \mathtt{OPT}'_{g_1}, \ldots, \mathtt{OPT}'_{g_d}$;

**for** $\beta \leftarrow 0$; $\beta \leq 1$; $\beta \leftarrow \beta + \frac{\varepsilon}{2}$ **do**

    Initialize $\alpha_{max} \leftarrow 1$, $\alpha_{min} \leftarrow 0$;

    **while** $\alpha_{max} - \alpha_{min} > \frac{\varepsilon}{2}$ **do**

        Set $\alpha \leftarrow (\alpha_{max} + \alpha_{min})/2$ and define $F'_{\alpha,\beta}(S)$ according to Eq. 3;

        $S \leftarrow \mathtt{CostEffectiveGreedy}(f, g_1, \ldots, g_d, c, k, \varepsilon)$ (for knapsack constraint) or $\mathtt{IterativeGreedy}(f, g_1, \ldots, g_d, \mathcal{I}(\mathcal{M}), \varepsilon)$ (for matroid constraint);

        **if** $F'_{\alpha,\beta}(S) \geq d + 1 - \frac{\varepsilon}{2}$ **then**

            $\alpha_{min} \leftarrow \alpha$ and $S_{\alpha,\beta} \leftarrow S$;

        **else**

            $\alpha_{max} \leftarrow \alpha$;

        **end**

    **end**

    Add $S_{\alpha_{min},\beta}$ to $\mathcal{S}$ and remove all $S_{\alpha',\beta'}$ with $\alpha' \leq \alpha_{min}$ and $\beta' < \beta$ from $\mathcal{S}$;

**end**

**return** $\mathcal{S}$;

**Function** $\mathtt{CostEffectiveGreedy}(f, g_1, \ldots, g_d, c, k, \varepsilon)$**:**

    Initialize $S \leftarrow \emptyset$;

    **while** $\exists v \in V \setminus S$ *such that* $c(S \cup \{v\}) \leq k(1 + \ln \frac{2d+2}{\varepsilon})$ **do**

        $I \leftarrow \{v \in V : c(S \cup \{v\}) \leq k(1 + \ln \frac{2d+2}{\varepsilon})\}$;

        $v^* \leftarrow \arg\max_{v \in I} \left(F'_{\alpha,\beta}(S \cup \{v\}) - F'_{\alpha,\beta}(S)\right)/c(v)$ and $S \leftarrow S \cup \{v^*\}$;

    **end**

    **return** $S$;

**Function** $\mathtt{IterativeGreedy}(f, g_1, \ldots, g_d, \mathcal{I}(\mathcal{M}), \varepsilon)$**:**

    **for** $l \leftarrow 1$; $l \leq 1 + \lceil \log_2 \frac{d+1}{\varepsilon} \rceil$; $l \leftarrow l + 1$ **do**

        $S_l \leftarrow \emptyset$;

        **while** $\exists v \in V : S_l \cup \{v\} \in \mathcal{I}(\mathcal{M})$ **do**

            $I \leftarrow \{v \in V : S_l \cup \{v\} \in \mathcal{I}(\mathcal{M})\}$;

            $v^* \leftarrow \arg\max_{v \in I} F'_{\alpha,\beta}(\cup_{j=1}^l S_j \cup \{v\}) - F'_{\alpha,\beta}(\cup_{j=1}^l S_j)$ and $S_l \leftarrow S_l \cup \{v^*\}$;

        **end**

    **end**

    **return** $S \leftarrow \bigcup_{l=1}^{1 + \lceil \log_2 \frac{d+1}{\varepsilon} \rceil} S_l$;

---

where the problem of division by zero is solved in the same way as for $F_{\alpha,\beta}$ when $\alpha$, $\beta$ or $\mathtt{OPT}'_f$, $\mathtt{OPT}'_{g_i}$ are equal to 0. Next, the following lemmas indicate that SMFR-DEC can still be answered approximately by solving the relaxed problem in Eq. 3.

**Lemma 1.** *Any set $S \in \mathcal{I}$ with $F'_{\alpha,\beta}(S) \geq d + 1 - \frac{\varepsilon}{2}$ must be a $(\delta\alpha - \frac{\varepsilon}{2}, \delta\beta - \frac{\varepsilon}{2})$-approximate solution to* SMFR*, where $\delta \in (0, 1 - 1/e]$ is the approximation factor of the algorithm used for* SMK *or* SMM*.*

**Lemma 2.** *If there is no set $S \in \mathcal{I}$ with $F'_{\alpha,\beta}(S) = d + 1$, no $(\alpha, \beta)$-approximate solution to* SMFR *exists.*

See Appendices A.1 and A.2 for the proofs of the above two lemmas.

Based on Lemmas 1 and 2, we propose SMFR-Saturate in Algorithm 1 for SMFR. Generally, SMFR-Saturate follows the same framework to handle the knapsack and matroid constraints but uses different greedy algorithms to obtain approximate solutions to SC on $F'_{\alpha,\beta}$. We first run an algorithm for SMK or SMM on each objective function individually with the same knapsack constraint $\mathcal{I}_k$ or matroid constraint $\mathcal{I}(\mathcal{M})$ to calculate $\texttt{OPT}'_f, \texttt{OPT}'_{g_1}, \ldots, \texttt{OPT}'_{g_d}$. Then, we iterate over each value of $\beta$ from 0 to 1 with an interval of $\frac{\varepsilon}{2}$. For each value of $\beta$, we perform a bisection search on $\alpha$ between 0 and 1. Given a pair of $\alpha$ and $\beta$, we formulate an instance of SC on $F'_{\alpha,\beta}$ in Eq. 3.

To address SC on $F'_{\alpha,\beta}$, we adopt two different types of greedy algorithms specific to the knapsack and matroid constraints, respectively. For a knapsack constraint $\mathcal{I}_k$, we run the $\texttt{CostEffectiveGreedy}$ algorithm, which starts from $S = \emptyset$ and adds the most "cost-effective" item $v^*$ with the largest ratio between its marginal gain w.r.t. $S$ and its cost $c(v^*)$ until no more items can be added with a relaxed knapsack constraint with a budget $k(1 + \ln \frac{2d+2}{\varepsilon})$, to find the candidate solution $S$. For a matroid constraint $\mathcal{I}(\mathcal{M})$, we run the $\texttt{IterativeGreedy}$ algorithm, which performs the classic greedy algorithm for SMM (Fisher et al., 1978) iteratively in $1 + \lceil \log_2 \frac{d+1}{\varepsilon} \rceil$ rounds. In the $l$-th round, we start from $S_l = \emptyset$ and add the item $v^*$ that satisfies $S_l \cup \{v^*\} \in \mathcal{I}(\mathcal{M})$ and has the largest marginal gain w.r.t. $\cup_{j=1}^l S_j$ until no more items can be added to $S_l$ under the knapsack constraint $\mathcal{I}(\mathcal{M})$. Finally, we return the union of the items selected over all rounds, i.e., $\bigcup_{l=1}^{1+\lceil \log_2 \frac{d+1}{\varepsilon} \rceil} S_l$, as the candidate solution $S$.

After computing a candidate solution $S$, if $F'_{\alpha,\beta}(S) \geq d + 1 - \frac{\varepsilon}{2}$, that is, $S$ reaches the "saturation level" w.r.t. $\alpha, \beta$ according to Lemma 1, we set $S$ as the current solution $S_{\alpha,\beta}$ and search in the upper half for a better solution with a higher value of $\alpha$; otherwise, we search in the lower half for a feasible solution. When $\alpha_{max} - \alpha_{min} \leq \frac{\varepsilon}{2}$, we add the solution $S_{\alpha_{min},\beta}$ to $\mathcal{S}$, remove all solutions dominated by $S_{\alpha_{min},\beta}$, and move on to the next value of $\beta$. Finally, all non-dominated solutions in $\mathcal{S}$ are returned for SMFR.

The theoretical guarantees of SMFR-Saturate for SMFR with knapsack and matroid constraints are analyzed in the following two theorems, respectively.

**Theorem 1.** *For* SMFR *with a knapsack constraint $\mathcal{I}_k$,* SMFR-Saturate *runs in $O(dt(\mathcal{A}) + \frac{n^2}{\varepsilon} \log \frac{1}{\varepsilon})$ time, where $t(\mathcal{A})$ is the time complexity of the $\delta$-approximation algorithm for* SMK*, and provides a set $\mathcal{S}$ of solutions with the following properties: (1) $|\mathcal{S}| = O(\frac{1}{\varepsilon})$, (2) $c(S) = O(k \log \frac{d}{\varepsilon})$ for each $S \in \mathcal{S}$, (3) for each $(\alpha^*, \beta^*)$-approximate Pareto optimal solution $S^*$ to* SMFR*, there must exist its corresponding solution $S \in \mathcal{S}$ such that $f(S) \geq (\delta\alpha^* - \varepsilon)\texttt{OPT}_f$ and $g_i(S) \geq (\delta\beta^* - \varepsilon)\texttt{OPT}_{g_i}, \forall i \in [d]$.*

**Theorem 2.** *For* SMFR *with a matroid constraint $\mathcal{I}(\mathcal{M})$,* SMFR-Saturate *runs in $O(dt(\mathcal{A}) + \frac{nr}{\varepsilon} \log^2 \frac{d}{\varepsilon})$ time, where $t(\mathcal{A})$ is the time complexity of the $\delta$-approximation algorithm for* SMM*, and provides a set $\mathcal{S}$ of solutions with the following properties: (1) $|\mathcal{S}| = O(\frac{1}{\varepsilon})$, (2) $|S| = O(r \log \frac{d}{\varepsilon})$ for each $S \in \mathcal{S}$, (3) for each $(\alpha^*, \beta^*)$-approximate Pareto optimal solution $S^*$ to* SMFR*, there must exist its corresponding solution $S \in \mathcal{S}$ such that $f(S) \geq (\delta\alpha^* - \varepsilon)\texttt{OPT}_f$ and $g_i(S) \geq (\delta\beta^* - \varepsilon)\texttt{OPT}_{g_i}, \forall i \in [d]$.*

See Appendices A.3 and A.4 for the proofs of the above two theorems.

## 5 Experiments

In this section, we present extensive experimental results to evaluate the performance of our proposed algorithm (SMFR-Saturate) on two benchmark problems, namely *Maximum Coverage* and *Recommendation*, using several real-world data sets. We compare SMFR-Saturate with the following non-trivial baselines.

- Greedy+Max (or Greedy): The original greedy algorithms for single-objective submodular maximization. For SMK, we adopt the $O(n^2)$-time Greedy+Max algorithm by Yaroslavtsev et al. (2020); and for SMM, we adopt the $O(nr)$-time Greedy algorithm by Fisher et al. (1978). Both algorithms have the same approximation factor of $1/2$.

- Saturate: The bicriteria approximation algorithms for the problem of *multi-objective submodular maximization* (MOSM) that maximizes the minimum among multiple (submodular) objective functions. As for SMFR, we should maximize the minimum among the $d + 1$ functions of $f$ and

$g_1, \ldots, g_d$. In particular, SATURATE for MOSM with knapsack and matroid constraints is presented in (Krause et al., 2008) and (Anari et al., 2019), respectively.

- SMSC: A $(0.16, 0.16)$-approximation algorithm for the problem of Submodular Maximization under Submodular Cover (SMSC) (Ohsaka & Matsuoka, 2021), which can be used for SMFR only when $d = 1$ by maximizing $f$ under the submodular cover constraint defined on $g_1$.

- BSM-SATURATE: The instance-dependent bicriteria approximation algorithm for balancing *utility* (i.e., maximizing $f$) and *fairness* (i.e., maximizing the minimum of $g_1, \ldots, g_d$) in (Wang et al., 2024).

- OPT: Formulating an instance of SMFR as an integer-linear program (ILP) and using a solver to enumerate its Pareto optimal solutions in the worst-case exponential time. The ILP formulations of SMFR for *Maximum Coverage* and *Recommendation* are deferred to Appendix B.

All algorithms are appropriately adapted to provide solutions without violating the specified constraints. We implemented them in Python 3, and for the OPT algorithm, we applied the Gurobi[3] optimizer to solve the ILP formulations of the *Maximum Coverage* and *Recommendation* instances. All algorithms except OPT were accelerated using the lazy-forward strategy (Leskovec et al., 2007), as this strategy cannot be applied to OPT. All experiments were run on a MacBook Pro laptop with an Apple M1 Max processor and 32GB memory running MacOS 14. Data and code are available publicly at `https://github.com/adrianfaz/Fair-Representation-in-Submodular-Subset-Selection-A-Pareto-Optimization-Approach`.

## 5.1 Maximum Coverage

**Setup.** In this subsection, we evaluate the performance of different algorithms for SMFR on the *Maximum Coverage* problem using two real-world data sets: *Facebook* and *DBLP*. The *Facebook* data set (Mislove et al., 2010) is an undirected graph of $1,216$ nodes and $42,443$ edges representing the friendships between Rice University students on Facebook, and the *DBLP* data set (Dong et al., 2023) is an undirected graph of $3,980$ nodes and $6,966$ edges denoting the coauthorships between researchers.

Our settings for *Maximum Coverage* follow those used in the existing literature on submodular maximization (Halabi et al., 2020; Ohsaka & Matsuoka, 2021; Wang et al., 2024). Given a graph $\mathcal{G} = (V, E)$, the utility (i.e., coverage) function is defined as $f(S) := |\bigcup_{v \in S} \mathcal{N}(v)|$, where $\mathcal{N}(v)$ is the set of nodes consisting of $v$ and its neighbors in $\mathcal{G}$. That is, the coverage of a set $S \subseteq V$ is measured by the number of nodes in the union of the neighborhoods of all nodes in $S$. To define the representativeness functions $g_1, g_2, \ldots, g_d$, we divide the node set into $d$ communities $C_1, \ldots, C_d$ such that $\bigcup_{i=1}^{d} C_i = V$. For each $i \in [d]$, the function $g_i$ is associated with a particular community $C_i$ as $g_i(S) := |\bigcup_{v \in S} \mathcal{N}(v) \cap C_i|$. That is, the representativeness of a set $S$ for a community $C_i$ is measured by the number of nodes in $C_i$ covered by $S$. For both data sets, the node set $V$ is partitioned into four disjoint groups using the Louvain method (Blondel et al., 2008) for community detection. We then index the four communities according to their sizes as $|C_1| \geq |C_2| \geq |C_3| \geq |C_4|$. For the *DBLP* data set, we follow the scheme of (Jin et al., 2021) to define a knapsack constraint by assigning a cost of 0.2 times its degree to each node and then normalizing all costs by the minimum cost. For the *Facebook* data set, we define a partition matroid constraint by dividing all nodes into 4 disjoint groups based on a sensitive attribute (i.e., *age*). We then follow the rule of *equal representation* (Halabi et al., 2020) to set the same upper bound $k \in \mathbb{Z}^+$ for each age group, resulting in a partition matroid of rank $r = 4k$.

**Results.** Figures 1a–1c and 2a–2c present the trade-offs between $\alpha$ and $\beta$ achieved by each algorithm for different instances of SMFR on *Maximum Coverage* with knapsack and matroid constraints on the *DBLP* and *Facebook* data sets, respectively. We fix $k = 40$ for the knapsack constraint and $k = 5$ (and thus $r = 20$) for the matroid constraint. We set $d = 1, 2$, and 4 by considering the representativeness functions on the first group $C_1$, the first two groups $C_1$ and $C_2$, and all four groups from $C_1$ to $C_4$. In each of these figures, the x- and y-axes represent the values of $\alpha$ and $\beta$ for all solutions with a distinct marker for each algorithm. Furthermore, we also use a black line and a red line to denote the optimal Pareto frontier returned by OPT and its approximation returned by SMFR-SATURATE. From the results, we observe that the Pareto frontiers provided by SMFR-SATURATE are equal or very close to the optimal ones. This confirms the effectiveness of

---

[3]`https://www.gurobi.com/solutions/gurobi-optimizer/`

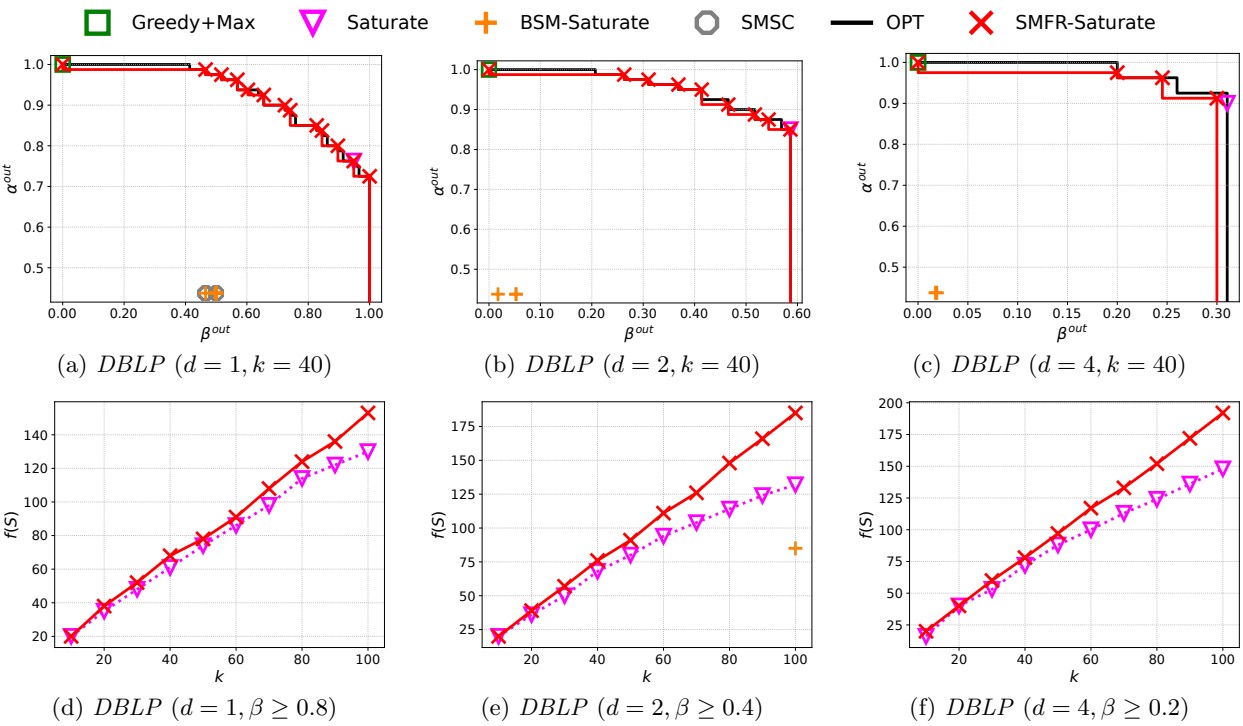

Figure 1: Results for *Maximum Coverage* on the *DBLP* data set, with knapsack constraints.

SMFR-Saturate for the SMFR problem. We also find that the Greedy+Max and Greedy algorithms, which focus solely on maximizing $f$, generally provide solutions with low values of $\beta$, indicating significant neglect of representativeness functions. Furthermore, Saturate, which maximizes the minimum among all representativeness and utility functions and does not allow any trade-off between $f$ and $g$ by design, in some cases (e.g., Figures 1a–1c), it provides a solution with the highest value of $\beta$ while having a value of $\alpha$ equal to or close to that of SMFR-Saturate and OPT for maximum $\beta$. However, it returns inferior solutions dominated by those of SMFR-Saturate in other cases. BSM-Saturate and SMSC provide different trade-offs between $f$ and $g$ by adjusting the threshold value $\tau$ in their definitions. The trade-offs reported by SMSC are marginally better than those of SMFR-Saturate on the *Facebook* data set with matroid constraints (Figure 2a). Conversely, it performs poorly for knapsack constraints (Figure 1a). In fact, SMSC is a special case of SMFR when $d = 1$, the matroid/knapsack constraint is reduced to the cardinality constraint, and the trade-off between $f$ and $g$ is predetermined by $\tau$. It is also noted that SMSC cannot work when $d > 1$. Although BSM-Saturate does not have the restriction of $d = 1$, its trade-offs are never better than those obtained by SMFR-Saturate, and significantly worse for *Maximum Coverage* on the *DBLP* data set with knapsack constraints (Figures 1a–1c).

Figures 1d–1f and 2d–2f report the effect of the parameter $k$, which directly decides the solution size, on the performance of each algorithm for different instances of SMFR in the context of *Maximum Coverage* with knapsack and matroid constraints on the *DBLP* and *Facebook* data sets, respectively. In each plot, the x-axis represents the value of $k$ in the knapsack or matroid constraint, and the y-axis represents the maximum utility value $f(S)$ among all solutions with a certain level of representativeness, i.e., the value of $\beta$ reaches a given threshold, provided by an algorithm. We also set $d = 1, 2$, and 4 by considering the representativeness functions on $C_1$, $C_1\&C_2$, and $C_1$–$C_4$. Only solutions with $\beta \geq 0.8$ are considered for $d = 1$, $\beta \geq 0.4$ for $d = 2$, and $\beta \geq 0.2$ for $d = 4$. A unique marker and a distinct line color are used for each algorithm. From Figures 1d–1f, we observe that the solutions provided by SMFR-Saturate consistently achieve the highest utility value $f(S)$ across all values of $k$ in the knapsack constraint. The absence of SMSC and BSM-Saturate indicates that they fail to provide solutions with an adequate level of representativeness (i.e., the value of $\beta$ is below the given thresholds), with the only exception shown in Figure 1e when $k = 100$.

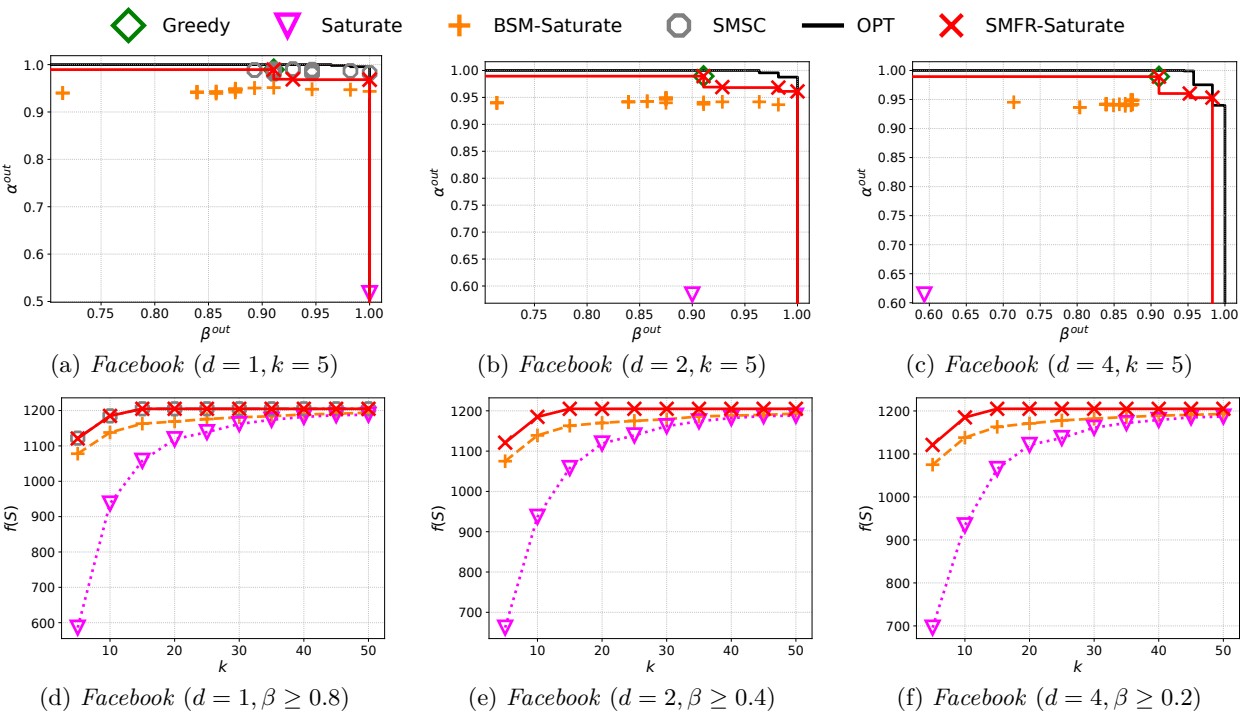

Figure 2: Results for *Maximum Coverage* on the *Facebook* data set, with matroid constraints.

Furthermore, although SATURATE provides valid solutions in all cases, the gap in the utility value $f(S)$ between SMFR-SATURATE and SATURATE widens as the knapsack restriction becomes less stringent (i.e., increasing $k$), for all values of the number of representativeness functions $d$. Figures 2d–2f show that across all values of $k$, the solutions provided by SMFR-SATURATE always achieve utility values $f(S)$ higher than those of BSM-SATURATE and SATURATE. Unlike the case of knapsack constraints, the gap in the utility value $f(S)$ among all methods decreases as the matroid constraint becomes less stringent (i.e. increasing $k$), for all values of the number of representativeness functions $d$. In the case of $d = 1$, SMSC and SMFR-SATURATE exhibit the same performance, as shown in Figure 2d. The above results confirm that when the trade-off level between $f$ and $g$ is pre-specified, one can still find a corresponding solution from those of SMFR-SATURATE that is comparable to or better than those provided by other baselines.

## 5.2 Recommendation

**Setup.** In this subsection, we evaluate the performance of different algorithms for SMFR on the *Recommendation* problem using another two real-world data sets: *X-Wines* (de Azambuja et al., 2023) and *MovieLens*[4]. The *X-Wines* data set consists of 150 000 ratings from 10 561 users on 1 007 wines, where each rating takes a value in the range $[1.0, 1.5, \ldots, 5.0]$. Moreover, each wine in the data set is associated with one or more food types that pair with the wine itself; we group these food types into four categories: *"meat"*, *"fish"*, *"pasta"*, and *"cheese"*. The *MovieLens* data set consists of 100 000 ratings from 600 users on 9 000 movies, where each rating takes a value in the range $[0.5, 1.0, \ldots, 5.0]$. Each movie in the data set is associated with one or more genres, with a total of 20 genres.

Our experimental settings are similar to those adopted in (Ohsaka & Matsuoka, 2021). In the following, we use the term *"item"* to refer to either a wine in the *X-Wines* data set or a movie in the *MovieLens* data set. By performing the non-negative matrix factorization[5] (NMF) on the user-item rating matrix with $p = 32$ factors, we obtain a 32-dimensional feature vector for each item and user. Denoting by $\mathbf{v}_i \in \mathbb{R}^p$ the feature vector of

---

[4]https://grouplens.org/datasets/movielens/

[5]https://scikit-learn.org/stable/modules/generated/sklearn.decomposition.NMF.html

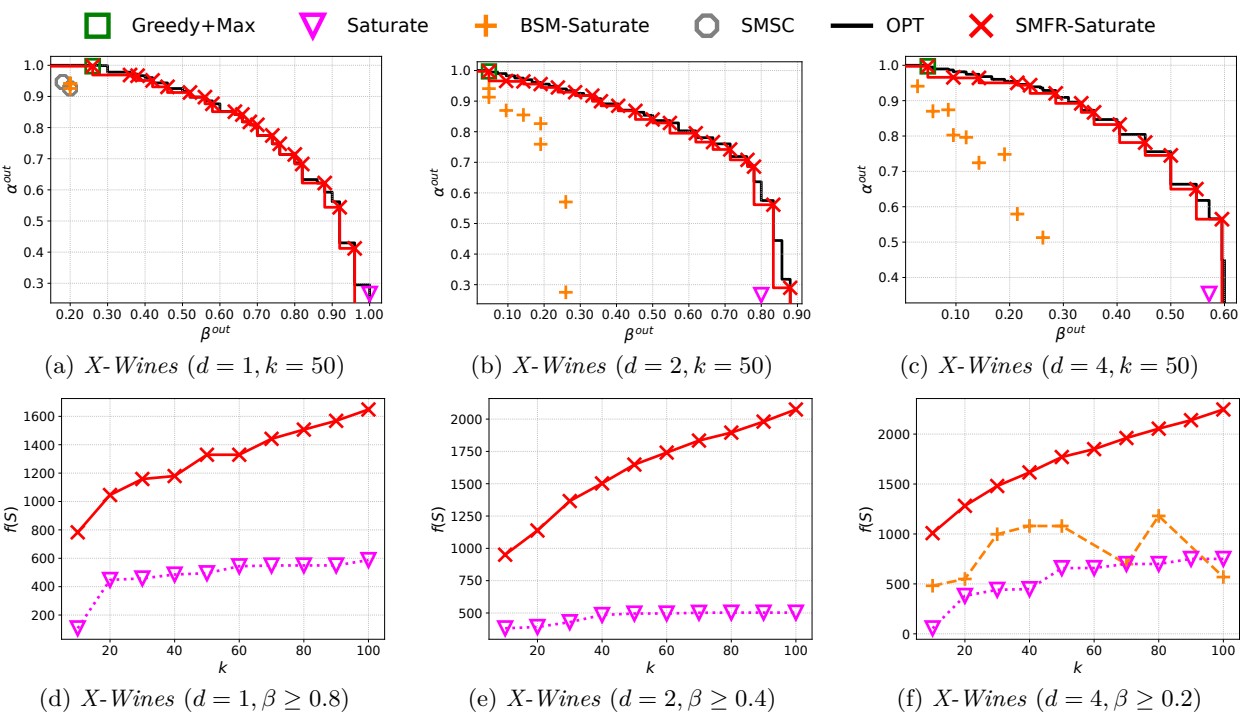

Figure 3: Results for *Recommendation* on the *X-Wines* data set, with knapsack constraints.

item $i$, and by $\mathbf{u}_j \in \mathbb{R}^p$ the feature vector of user $j$, the inner product $\langle \mathbf{v}_i, \mathbf{v}_j \rangle$ between two feature vectors associated with two items measures their similarity. The same holds for users and items as well: $\langle \mathbf{v}_i, \mathbf{u}_j \rangle$ indicates the level at which a user likes an item. To design the utility function $f$ according to the facility location objective, we select a subset $T$ of items with at least 54 ratings ($|T| = 503$ for the *X-Wines* data set, and $|T| = 403$ for the *MovieLens* data set), and define $f : 2^V \to \mathbb{R}^+$ as $f(S) := \sum_{t \in T} \max_{s \in S} \langle \mathbf{v}_s, \mathbf{v}_t \rangle$, where $V$ is the set of all items in each data set: $|V| = 1\,007$ for the *X-Wines* data set, and $|V| = 9\,000$ for the *MovieLens* data set. The function $f$ captures how well the selected subset $S$ can represent all items in $T$ in the sense that for any item $t \in T$, there exists an item in $S$ that is highly similar to it. This function, as defined, is known to be monotone and submodular (Frieze, 1974). To define the representativeness functions $g_1, g_2, \ldots, g_d$, we consider using, for the *X-Wines* data set, the food type categories with which a wine pair, and, for the *MovieLens* data set, the genres to which a movie belongs. Specifically, for the *X-Wines* data set, we divide wines into four groups according to their associated food type categories as $G_1$ (*meat*), $G_2$ (*fish*), $G_3$ (*pasta*), and $G_4$ (*cheese*). Similarly, for the *MovieLens* data set, we divide movies into four groups according to their genres as $G_1$ (*dramas*), $G_2$ (*comedies*), $G_3$ (*thrillers*), and $G_4$ (*action movies*). Then, each $g_i$ function is associated with a particular set of items and is defined as $g_i(S) := |S \cap G_i|$. To be specific, the representativeness of $S$ for $G_i$ is measured by the number of items in $S$ selected from $G_i$. For the *X-Wines* data set, we define a knapsack constraint by assigning to each item (wine) a random integer cost in the range $[1, 10]$. For the *MovieLens* data set, to define a matroid constraint, we partition the movies into 7 groups according to their release dates: $[1900, 1950)$, $[1950, 1970)$, $[1970, 1980)$, $[1980, 1990)$, $[1990, 2000)$, $[2000, 2010)$, and $[2010, 2019)$. We also use an equal upper bound $k \in \mathbb{Z}^+$ for each group, resulting in a partition matroid of rank $r = 7k$.

**Results.** Figures 3 and 4 present the performance of each algorithm for different instances of SMFR on *Recommendation* with knapsack and matroid constraints on the *X-Wines* and *MovieLens* data sets, respectively. In general, we observe results similar to those for *Maximum Coverage* and further confirm the effectiveness of SMFR-SATURATE for SMFR in different applications. The absence of OPT in Figures 4a–4c is due to the inefficiency of the ILP solver: it cannot finish on any SMFR instance for the *MovieLens* data set within one hour. We also find that SMFR-SATURATE shows more significant advantages over SMSC

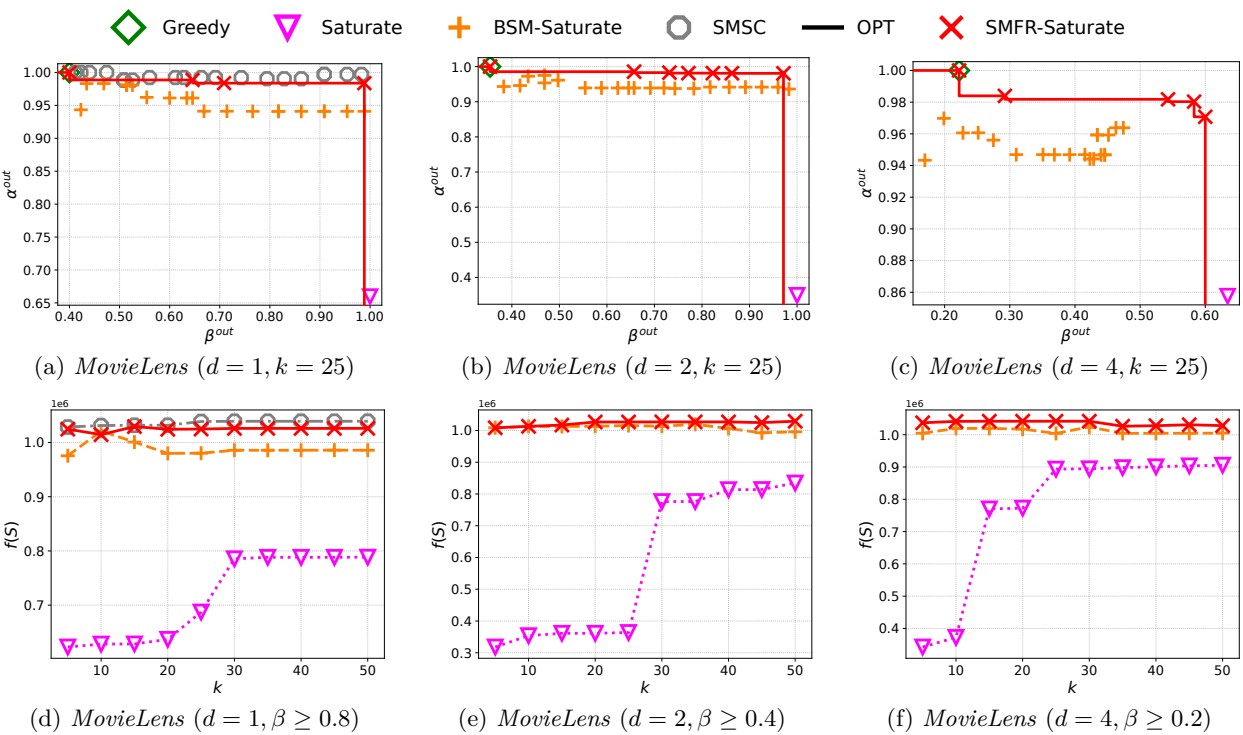

Figure 4: Results for *Recommendation* on the *MovieLens* data set, with matroid constraints.

and BSM-SATURATE for the knapsack constraints than for the matroid constraints. In particular, SMSC slightly outperforms SMFR-SATURATE when $d = 1$ on the *MovieLens* data set, with matroid constraints. This is because the solutions with cardinality constraints are typically very close to those with the partition matroid constraints that we define but differ significantly from those with knapsack constraints. As such, SMSC, which is designed specifically for cardinality constraints, can achieve good performance under matroid constraints without adaptations. Again, SMSC is not comparable to SMFR-SATURATE in other cases.

Finally, we omit the remaining experimental results due to space limitations. Please refer to Appendix C for those results, which further confirm the effectiveness of SMFR-SATURATE in other experimental settings and provide additional evaluations for the efficiency of SMFR-SATURATE and other baselines.

## 6  Conclusion and Future Work

In this paper, we study a novel multi-objective combinatorial optimization problem called *Submodular Maximization with Fair Representation* (SMFR), which aims to select subsets from a ground set under a specific knapsack or matroid constraint such that a submodular (utility) function $f$ is maximized while $d$ submodular (representativeness) functions $g_1, \ldots, g_d$ are also maximized. We show the hardness of finding optimal solutions to SMFR and propose a Pareto optimization approach, SMFR-SATURATE, to enumerating a set of approximate solutions to all Pareto optimal solutions with different trade-offs between multiple objectives for SMFR. Finally, we demonstrate the effectiveness of SMFR-SATURATE in two classic submodular problems, *Maximum Coverage* and *Recommendation*, using real-world data.

We note that SMFR-SATURATE still has several limitations. For example, it cannot support more general classes of functions in subset selection problems, such as non-monotone and weakly submodular functions, and more complex constraints, including the intersection of multiple knapsack and matroid constraints and the $P$-system constraint. We would like to extend SMFR-SATURATE to support them in future work. Furthermore, it would also be interesting to expand the realm of *fair submodular optimization* (Halabi et al., 2023; Mehrotra & Vishnoi, 2023) by considering more novel and practical notions of fairness.

## Acknowledgments

Yanhao Wang was supported by the National Natural Science Foundation of China under grant number 62202169 and the start-up funding from ECNU.

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

# A    Proofs of Lemmas and Theorems

## A.1    Proof of Lemma 1

**Lemma 1.** *Any set $S \in \mathcal{I}$ with $F'_{\alpha,\beta}(S) \geq d+1-\frac{\varepsilon}{2}$ must be a $(\delta\alpha - \frac{\varepsilon}{2}, \delta\beta - \frac{\varepsilon}{2})$-approximate solution to* SMFR, *where $\delta \in (0, 1-1/e]$ is the approximation factor of the algorithm used for* SMK *or* SMM.

*Proof.* We first consider the two special cases of $\alpha = 0$ and $\beta = 0$. When $\alpha = 0$ or $\beta = 0$, if $F'_{\alpha,\beta}(S) > d+1-\frac{\varepsilon}{2}$, we will have $\frac{g_i(S)}{\beta \mathrm{OPT}'_{g_i}} > 1 - \frac{\varepsilon}{2}$ for every $i \in [d]$ or $\frac{f(S)}{\alpha \mathrm{OPT}'_f} > 1 - \frac{\varepsilon}{2}$. In the general case of $\alpha, \beta > 0$, if $F'_{\alpha,\beta}(S) > d+1-\frac{\varepsilon}{2}$, we will have $\frac{f(S)}{\alpha \mathrm{OPT}'_f} > 1 - \frac{\varepsilon}{2}$ and $\frac{g_i(S)}{\beta \mathrm{OPT}'_{g_i}} > 1 - \frac{\varepsilon}{2}$ for every $i \in [d]$ at the same time. Thus, it holds that

$$f(S) \geq (1 - \frac{\varepsilon}{2})\alpha\mathrm{OPT}'_f \geq \delta\alpha(1 - \frac{\varepsilon}{2})\mathrm{OPT}_f \geq (\delta\alpha - \frac{\varepsilon}{2})\mathrm{OPT}_f$$

and

$$g_i(S) \geq (1 - \frac{\varepsilon}{2})\beta\mathrm{OPT}'_{g_i} \geq \delta\beta(1 - \frac{\varepsilon}{2})\mathrm{OPT}_{g_i} \geq (\delta\beta - \frac{\varepsilon}{2})\mathrm{OPT}_{g_i}, \forall i \in [d].$$

Therefore, $S$ is a $(\delta\alpha - \frac{\varepsilon}{2}, \delta\beta - \frac{\varepsilon}{2})$-approximate solution to SMFR. $\qquad\square$

## A.2    Proof of Lemma 2

**Lemma 2.** *If there is no set $S \in \mathcal{I}$ with $F'_{\alpha,\beta}(S) = d+1$, no $(\alpha, \beta)$-approximate solution to* SMFR *exists.*

*Proof.* If $F'_{\alpha,\beta}(S) < d+1$, then we will have $f(S) < \alpha\mathrm{OPT}'_f \leq \alpha\mathrm{OPT}_f$ or there is some $i \in [d]$ with $g_i(S) < \beta\mathrm{OPT}'_{g_i} \leq \beta\mathrm{OPT}_{g_i}$. Therefore, if $F'_{\alpha,\beta}(S) < d+1$, $S$ will not be an $(\alpha, \beta)$-approximate solution to SMFR. And if there is no set $S \in \mathcal{I}$ with $F'_{\alpha,\beta}(S) = d+1$, no $(\alpha, \beta)$-approximate solution to SMFR exists. $\qquad\square$

## A.3    Proof of Theorem 1

**Theorem 1.** *For* SMFR *with a knapsack constraint $\mathcal{I}_k$,* SMFR-SATURATE *runs in $O(dt(\mathcal{A}) + \frac{n^2}{\varepsilon}\log\frac{1}{\varepsilon})$ time, where $t(\mathcal{A})$ is the time complexity of the $\delta$-approximation algorithm for* SMK, *and provides a set $\mathcal{S}$ of solutions with the following properties: (1) $|\mathcal{S}| = O(\frac{1}{\varepsilon})$, (2) $c(S) = O(k\log\frac{d}{\varepsilon})$ for each $S \in \mathcal{S}$, (3) for each $(\alpha^*, \beta^*)$-approximate Pareto optimal solution $S^*$ to* SMFR, *there must exist its corresponding solution $S \in \mathcal{S}$ such that $f(S) \geq (\delta\alpha^* - \varepsilon)\mathrm{OPT}_f$ and $g_i(S) \geq (\delta\beta^* - \varepsilon)\mathrm{OPT}_{g_i}, \forall i \in [d]$.*

*Proof.* Let us first analyze the time complexity of SMFR-SATURATE for a knapsack constraint $\mathcal{I}_k$. First, it runs the SMK algorithm $d+1$ times to compute $\mathrm{OPT}'_f$ and $\mathrm{OPT}'_{g_i}$ for every $i \in [d]$. Then, it iterates over $\lceil\frac{2}{\varepsilon}\rceil$ values of $\beta$ in the `for` loop. For each value of $\beta$, it attempts to use $O(\log\frac{1}{\varepsilon})$ different values of $\alpha$ in the bisection search. Finally, the subroutine `CostEffectiveGreedy` takes $O(n^2)$ time for SC on each $F'_{\alpha,\beta}$. In summary, the time complexity of SMFR-SATURATE for a knapsack constraint $\mathcal{I}_k$ is $O(dt(\mathcal{A}) + \frac{n^2}{\varepsilon}\log\frac{1}{\varepsilon})$ time, where $t(\mathcal{A})$ is the time complexity of the SMK algorithm.

For the solution $\mathcal{S}$ of SMFR-SATURATE, it is easy to see that $|\mathcal{S}| \leq \lceil\frac{2}{\varepsilon}\rceil$ and thus $|\mathcal{S}| = O(\frac{1}{\varepsilon})$ because SMFR-SATURATE adds at most one set to $\mathcal{S}$ for each value of $\beta$. Then, due to the condition in the `while` loop of the subroutine `CostEffectiveGreedy`, it must hold that $c(S) \leq k(1 + \ln\frac{2d+2}{\varepsilon})$ and thus $c(S) = O(k\log\frac{d}{\varepsilon})$ for each $S \in \mathcal{S}$. Finally, given an $(\alpha^*, \beta^*)$-approximate Pareto optimal solution $S^*$, there must exist a value of $\beta$ in the `for` loop such that $0 \leq \beta^* - \beta \leq \frac{\varepsilon}{2}$. Let $S_{\alpha_{min},\beta}$ be the solution of SMFR-SATURATE w.r.t. such $\beta$ and its corresponding $\alpha_{min}$. Since $F'_{\alpha_{min},\beta}(S_{\alpha_{min},\beta}) \geq d+1-\frac{\varepsilon}{2}$, $S_{\alpha_{min},\beta}$ is a $(\delta\alpha_{min} - \frac{\varepsilon}{2}, \delta\beta - \frac{\varepsilon}{2})$-approximate solution according to Lemma 1. Furthermore, we have $F'_{\alpha_{max},\beta}(S_{gr}) < d+1-\frac{\varepsilon}{2}$, where $S_{gr}$ is the solution w.r.t. $F'_{\alpha_{max},\beta}$ with a relaxed knapsack constraint for a budget $k(1 + \ln\frac{2d+2}{\varepsilon})$ returned by the subroutine `CostEffectiveGreedy` in Algorithm 1, and $\alpha_{max} - \alpha_{min} < \frac{\varepsilon}{2}$. Suppose that $S'_{gr}$ is the first intermediate subset of $S_{gr}$ with $c(S'_{gr}) \geq k\ln\frac{2d+2}{\varepsilon}$ constructed using the cost-effective greedy procedure. Let $S^*_k =$

$\arg \max_{S \in \mathcal{I}_k} F'_{\alpha_{max}, \beta}(S)$ and $\text{OPT}_{F'_{\alpha_{max}, \beta}} = F'_{\alpha_{max}, \beta}(S_k^*)$. According to the monotonicity and submodularity of $F'_{\alpha_{max}, \beta}$, we have

$$F'_{\alpha_{max}, \beta}(S_k^*) \le F'_{\alpha_{max}, \beta}(S_{gr}^{(i)}) + \sum_{v \in S_k^* \setminus S_{gr}^{(i)}} \Delta(v | S_{gr}^{(i)}) = F'_{\alpha_{max}, \beta}(S_{gr}^{(i)}) + \sum_{v \in S_k^* \setminus S_{gr}^{(i)}} \frac{c(v) \cdot \Delta(v | S_{gr}^{(i)})}{c(v)},$$

for any $S_{gr}^{(i)} \subset S_{gr}'$ after $i$ iterations and $\Delta(v | S_{gr}^{(i)}) = F'_{\alpha_{max}, \beta}(S_{gr}^{(i)} \cup \{v\}) - F'_{\alpha_{max}, \beta}(S_{gr}^{(i)})$. Let $u_i^*$ be the $i$-th item added to $S_{gr}'$ for any $i = 1, \ldots, |S_{gr}'|$. Based on the cost-effective greedy selection in Algorithm 1,

$$\frac{\Delta(u_{i+1}^* | S_{gr}^{(i)})}{c(u_{i+1}^*)} \ge \frac{\Delta(v | S_{gr}^{(i)})}{c(v)}$$

for any $v \in S_k^* \setminus S_{gr}^{(i)}$ and $i \in [0, \ldots, |S_{gr}' - 1|]$ because $c(v) \le k$ for any $v \in S_k^*$ and thus no item from $S_k^*$ is excluded from consideration due to budget violation when $u_{i+1}^*$ is added to $S_{gr}^{(i)}$. Therefore, we further obtain

$$F'_{\alpha_{max}, \beta}(S_k^*) \le F'_{\alpha_{max}, \beta}(S_{gr}^{(i)}) + \frac{\Delta(u_{i+1}^* | S_{gr}^{(i)})}{c(u_{i+1}^*)} \sum_{v \in S_k^* \setminus S_{gr}^{(i)}} c(v) \le F'_{\alpha_{max}, \beta}(S_{gr}^{(i)}) + \frac{\Delta(u_{i+1}^* | S_{gr}^{(i)})}{c(u_{i+1}^*)} \cdot k,$$

After rearranging the inequality above, we have

$$F'_{\alpha_{max}, \beta}(S_k^*) - F'_{\alpha_{max}, \beta}(S_{gr}^{(i+1)}) \le \left(1 - \frac{c(u_{i+1}^*)}{k}\right) \left(F'_{\alpha_{max}, \beta}(S_k^*) - F'_{\alpha_{max}, \beta}(S_{gr}^{(i)})\right).$$

Moreover, since $1 - x \le e^{-x}$ for any $x > 0$, it holds that $1 - \frac{c(u_{i+1}^*)}{k} \le \exp(-\frac{c(u_{i+1}^*)}{k})$. Therefore,

$$F'_{\alpha_{max}, \beta}(S_k^*) - F'_{\alpha_{max}, \beta}(S_{gr}^{(i+1)}) \le \exp(-\frac{c(u_{i+1}^*)}{k}) \cdot \left(F'_{\alpha_{max}, \beta}(S_k^*) - F'_{\alpha_{max}, \beta}(S_{gr}^{(i)})\right). \tag{4}$$

By applying Eq. 4 recursively to $i = 0, \ldots |S_{gr}'| - 1$, we have

$$\begin{aligned}
F'_{\alpha_{max}, \beta}(S_k^*) - F'_{\alpha_{max}, \beta}(S_{gr}') &\le \exp(-\frac{c(u_{i+1}^*)}{k}) \cdot \left(F'_{\alpha_{max}, \beta}(S_k^*) - F'_{\alpha_{max}, \beta}(S_{gr}^{(i)})\right) \\
&\le \exp(-\frac{c(u_{i+1}^*)}{k}) \exp(-\frac{c(u_i^*)}{k}) \left(F'_{\alpha_{max}, \beta}(S_k^*) - F'_{\alpha_{max}, \beta}(S_{gr}^{(i-1)})\right) \\
&\le \ldots \ldots \le \exp(-\frac{\sum_{i=0}^{|S_{gr}'|-1} c(u_{i+1}^*)}{k}) F'_{\alpha_{max}, \beta}(S_k^*) \\
&= \exp(-\frac{c(S_{gr}')}{k}) F'_{\alpha_{max}, \beta}(S_k^*) = \exp(-\frac{c(S_{gr}')}{k}) \text{OPT}_{F'_{\alpha_{max}, \beta}}.
\end{aligned}$$

Since $c(S_{gr}') \ge k \ln \frac{2d+2}{\varepsilon}$, it holds that

$$F'_{\alpha_{max}, \beta}(S_{gr}') \ge (1 - \exp\left(-\frac{c(S_{gr}')}{k}\right)) \text{OPT}_{F'_{\alpha_{max}, \beta}} \ge (1 - \frac{\varepsilon}{2d+2}) \text{OPT}_{F'_{\alpha_{max}, \beta}}.$$

In addition, $F'_{\alpha_{max}, \beta}(S_{gr}) \ge F'_{\alpha_{max}, \beta}(S_{gr}')$ since $S_{gr}' \subseteq S_{gr}$. Therefore, we have $\text{OPT}_{F'_{\alpha_{max}, \beta}} < d + 1$ and, according to Lemma 1, there does not exist any $(\alpha_{max}, \beta)$-approximate solution of cost at most $k$. Since $S^*$ is an $(\alpha^*, \beta^*)$-approximate Pareto optimal solution and $\beta \le \beta^*$, $S^*$ must be an $(\alpha^*, \beta)$-approximate solution of cost at most $k$. As such, we obtain $\alpha_{max} > \alpha^*$ and $\alpha_{min} > \alpha^* - \frac{\varepsilon}{2}$. Because we have shown that $S_{\alpha_{min}, \beta}$ is a $(\delta \alpha_{min} - \frac{\varepsilon}{2}, \delta \beta - \frac{\varepsilon}{2})$-approximate solution, $S_{\alpha_{min}, \beta}$ is guaranteed to be a $(\delta \alpha^* - \varepsilon, \delta \beta^* - \varepsilon)$-approximate solution. If $S_{\alpha_{min}, \beta}$ is included in $\mathcal{S}$, we will conclude the proof directly; otherwise, the solution in $\mathcal{S}$ dominating $S_{\alpha_{min}, \beta}$ can confirm our conclusion. $\qquad \square$

### A.4 Proof of Theorem 2

**Theorem 2.** *For* SMFR *with a matroid constraint* $\mathcal{I}(\mathcal{M})$, SMFR-SATURATE *runs in* $O(dt(\mathcal{A}) + \frac{nr}{\varepsilon} \log^2 \frac{d}{\varepsilon})$ *time, where* $t(\mathcal{A})$ *is the time complexity of the $\delta$-approximation algorithm for* SMM*, and provides a set* $\mathcal{S}$ *of solutions with the following properties: (1)* $|\mathcal{S}| = O(\frac{1}{\varepsilon})$, *(2)* $|S| = O(r \log \frac{d}{\varepsilon})$ *for each* $S \in \mathcal{S}$, *(3) for each* $(\alpha^*, \beta^*)$-*approximate Pareto optimal solution* $S^*$ *to* SMFR*, there must exist its corresponding solution* $S \in \mathcal{S}$ *such that* $f(S) \geq (\delta\alpha^* - \varepsilon)\mathtt{OPT}_f$ *and* $g_i(S) \geq (\delta\beta^* - \varepsilon)\mathtt{OPT}_{g_i}, \forall i \in [d]$.

*Proof.* Let us analyze the time complexity of SMFR-SATURATE for a matroid constraint $\mathcal{I}(\mathcal{M})$. First, it runs the SMM algorithm $d + 1$ times to compute $\mathtt{OPT}'_f$ and $\mathtt{OPT}'_{g_i}$ for every $i \in [d]$. Then, it iterates over $\lceil \frac{2}{\varepsilon} \rceil$ values of $\beta$ in the for loop. For each value of $\beta$, it attempts to use $O(\log \frac{1}{\varepsilon})$ different values of $\alpha$ in the bisection search. Finally, the subroutine IterativeGreedy takes $O(nr)$ time per round and runs in $O(\log \frac{d}{\varepsilon})$ rounds. In summary, the time complexity of SMFR-SATURATE for a matroid constraint $\mathcal{I}(\mathcal{M})$ is $O(dt(\mathcal{A}) + \frac{nr}{\varepsilon} \log \frac{d}{\varepsilon} \log \frac{1}{\varepsilon})$ time, where $t(\mathcal{A})$ is the time complexity of the SMM algorithm, and can be simplified as $O(dt(\mathcal{A}) + \frac{nr}{\varepsilon} \log^2 \frac{d}{\varepsilon})$.

For the solution $\mathcal{S}$ of SMFR-SATURATE, it is easy to see that $|\mathcal{S}| \leq \lceil \frac{2}{\varepsilon} \rceil$ and thus $|\mathcal{S}| = O(\frac{1}{\varepsilon})$ because SMFR-SATURATE adds at most one set to $\mathcal{S}$ for each value of $\beta$. Then, because the subroutine IterativeGreedy runs in at most $1 + \lceil \log_2 \frac{d+1}{\varepsilon} \rceil$ rounds and the size of each $S_l$ is bounded by the rank $r$ of the matroid $\mathcal{M}$, it must hold that $|S| \leq r \cdot (1 + \lceil \log_2 \frac{d+1}{\varepsilon} \rceil)$ and thus $|S| = O(r \log \frac{d}{\varepsilon})$ for each $S \in \mathcal{S}$. Finally, given an $(\alpha^*, \beta^*)$-approximate Pareto optimal solution $S^*$, there must exist a value of $\beta$ in the for loop such that $0 \leq \beta^* - \beta \leq \frac{\varepsilon}{2}$. Let $S_{\alpha_{min},\beta}$ be the solution of SMFR-SATURATE w.r.t. such $\beta$ and its corresponding $\alpha_{min}$. Since $F'_{\alpha_{min},\beta}(S_{\alpha_{min},\beta}) \geq d + 1 - \frac{\varepsilon}{2}$, $S_{\alpha_{min},\beta}$ is a $(\delta\alpha_{min} - \frac{\varepsilon}{2}, \delta\beta - \frac{\varepsilon}{2})$-approximate solution according to Lemma 1. Furthermore, we have $F'_{\alpha_{max},\beta}(S_{gr}) < d + 1 - \frac{\varepsilon}{2}$, where $S_{gr}$ is the solution w.r.t. $F'_{\alpha_{max},\beta}$ returned by the subroutine IterativeGreedy in Algorithm 1, and $\alpha_{max} - \alpha_{min} < \frac{\varepsilon}{2}$. Since IterativeGreedy runs a $\frac{1}{2}$-approximation greedy algorithm for submodular maximization with matroid constraints in each round, we have

$$F'_{\alpha_{max},\beta}(S_1) - F'_{\alpha_{max},\beta}(\emptyset) \geq \left(1 - \frac{1}{2}\right) \cdot \max_{S' \in \mathcal{I}(\mathcal{M})} (F'_{\alpha_{max},\beta}(S') - F'_{\alpha_{max},\beta}(\emptyset)).$$

Since $\widetilde{f}(S) = f(S \cup A) - f(S)$ is nonnegative, monotone, and submodular if $f(\cdot)$ is nonnegative, monotone, and submodular for any $A \subseteq V$, we can extend the above result for each round $l > 1$ as follows:

$$F'_{\alpha_{max},\beta}(\cup_{j=1}^l S_j) - F'_{\alpha_{max},\beta}(\cup_{j=1}^{l-1} S_j) \geq \left(1 - \frac{1}{2}\right) \cdot \max_{S'_l \in \mathcal{I}(\mathcal{M})} (F'_{\alpha_{max},\beta}(S'_l \cup (\cup_{j=1}^{l-1} S_j)) - F'_{\alpha_{max},\beta}(\cup_{j=1}^{l-1} S_j))$$

$$\geq \left(1 - \frac{1}{2}\right) \cdot \max_{S' \in \mathcal{I}(\mathcal{M})} (F'_{\alpha_{max},\beta}(S') - F'_{\alpha_{max},\beta}(\cup_{j=1}^{l-1} S_j)).$$

By induction, we obtain the following:

$$F'_{\alpha_{max},\beta}(\cup_{j=1}^l S_j) \geq \left(1 - \frac{1}{2^l}\right) \cdot \max_{S' \in \mathcal{I}(\mathcal{M})} F'_{\alpha_{max},\beta}(S') = \left(1 - \frac{1}{2^l}\right)\mathtt{OPT}_{F'_{\alpha_{max},\beta}}.$$

Since $S_{gr} = \cup_{j=1}^{1+\lceil \log_2 \frac{d+1}{\varepsilon} \rceil} S_j$, we have

$$F'_{\alpha_{max},\beta}(S_{gr}) \geq \left(1 - \frac{1}{2^l}\right) \cdot \mathtt{OPT}_{F'_{\alpha_{max},\beta}} \geq \left(1 - \frac{\varepsilon}{2d+2}\right) \cdot \mathtt{OPT}_{F'_{\alpha_{max},\beta}}.$$

Therefore, we have $\mathtt{OPT}_{F'_{\alpha_{max},\beta}} < d + 1$ and, according to Lemma 1, there does not exist any $(\alpha_{max}, \beta)$-approximate solution under matroid constraint $\mathcal{I}(\mathcal{M})$. Since $S^*$ is an $(\alpha^*, \beta^*)$-approximate Pareto optimal solution and $\beta \leq \beta^*$, $S^*$ must be an $(\alpha^*, \beta)$-approximate solution under matroid constraint $\mathcal{I}(\mathcal{M})$. As such, we obtain $\alpha_{max} > \alpha^*$ and $\alpha_{min} > \alpha^* - \frac{\varepsilon}{2}$. Because we have shown that $S_{\alpha_{min},\beta}$ is a $(\delta\alpha_{min} - \frac{\varepsilon}{2}, \delta\beta - \frac{\varepsilon}{2})$-approximate solution, $S_{\alpha_{min},\beta}$ is guaranteed to be a $(\delta\alpha^* - \varepsilon, \delta\beta^* - \varepsilon)$-approximate solution. If $S_{\alpha_{min},\beta}$ is included in $\mathcal{S}$, we will conclude the proof directly; otherwise, the solution in $\mathcal{S}$ dominating $S_{\alpha_{min},\beta}$ can confirm our conclusion. $\qquad\square$

## B  ILP Formulations

In this section, we present the integer linear programming (ILP) formulations for the *Maximum Coverage* and *Recommendation* problems, specifically tailored to the SMFR problem, as defined in Section 5.1 and Section 5.2, respectively. Any ILP solver can be employed to identify optimal solutions for small SMFR instances on *Maximum Coverage* and *Recommendation*. For our experimental results in Section 5 and Appendix C, we refer to this approach as the OPT algorithm. Note that these formulations are specifically designed for these settings and cannot be applied directly to general SMFR problems.

Problems 5 and 6 are specialized versions of the standard ILP formulation of SMFR on *Maximum Coverage*[6] in Section 5.1, with knapsack and partition matroid constraints, respectively.

$$\max \quad \sum_{j\in[m]} y_j \qquad (5)$$
$$\text{subject to} \quad \sum_{l\in[n]} c_l x_l \le k$$
$$\sum_{e_j\in S_l} x_l \ge y_j, \qquad \forall j\in[m]$$
$$\sum_{e_j\in C_i} y_j \ge \beta\,\mathtt{OPT}_{g_i}, \qquad \forall i\in[d]$$
$$y_j \in \{0,1\}, \qquad \forall j\in[m]$$
$$x_l \in \{0,1\}, \qquad \forall l\in[n]$$

$$\max \quad \sum_{j\in[m]} y_j \qquad (6)$$
$$\text{subject to} \quad \sum_{S_l\in V_t} x_l \le k, \qquad \forall t\in[p]$$
$$\sum_{e_j\in S_l} x_l \ge y_j, \qquad \forall j\in[m]$$
$$\sum_{e_j\in C_i} y_j \ge \beta\,\mathtt{OPT}_{g_i}, \qquad \forall i\in[d]$$
$$y_j \in \{0,1\}, \qquad \forall j\in[m]$$
$$x_l \in \{0,1\}, \qquad \forall l\in[n]$$

These ILPs maximize the *coverage* (i.e., the utility function $f$ in SMFR) on a universe $U=\{e_1,\ldots,e_m\}$ of $m$ elements and a collection $V=\{S_1,\ldots,S_n\}$ of $n$ sets ($S_l\subseteq V,\forall l\in[n]$), subject to additional coverage constraints on each subset $C_1,\ldots,C_d$ of $U$ (w.r.t. each representativeness function $g_1,\ldots,g_d$ in SMFR). In both formulations, $x_l$ indicates whether $S_l\in V$ is included in the solution $S$, and $y_j$ indicates whether $e_j\in U$ is covered by $S$. Problem 5 is specific to the knapsack constraint defined on a budget $k\in\mathbb{Z}^+$ and a cost function $c(\cdot)$. Problem 6 is specific to the partition matroid constraint, where $V$ is divided into $p$ disjoint partitions $V_1,\ldots,V_p$ and at most $k$ sets can be selected from each partition. Solving optimally Problems 5 and 6 with $\beta=0$ and $U=C_i$ yields the value of $\mathtt{OPT}_{g_i}$ for each representativeness function $g_i$ corresponding to the knapsack and the partition matroid constraints, respectively.

Problems 7 and 8 are specialized versions of the ILP formulation for capacitated facility location[7], with a benefit matrix $B=\{b_{jl}=\langle\mathbf{v}_i,\mathbf{v}_j\rangle : j\in[m], l\in[n]\}\in\mathbb{R}^{m\times n}$ ($m=|T|$ and $n=|V|$), specifically designed for SMFR on the *Recommendation* setting in Section 5.2, with knapsack and partition matroid constraints, respectively.

$$\max \quad \sum_{j\in[m]}\sum_{l\in[n]} b_{jl} y_{jl} \qquad (7)$$
$$\text{subject to} \quad \sum_{l\in[n]} c_l x_l \le k$$
$$\sum_{l\in[n]} y_{jl} \le 1, \qquad \forall j\in[m]$$
$$y_{jl} \le x_l, \qquad \forall j\in[m], l\in[n]$$
$$\sum_{e_l\in C_i} x_l \ge \beta\,\mathtt{OPT}_{g_i}, \qquad \forall i\in[d]$$
$$y_{jl} \in \{0,1\}, \qquad \forall j\in[m], l\in[n]$$
$$x_l \in \{0,1\}, \qquad \forall l\in[n]$$

$$\max \quad \sum_{j\in[m]}\sum_{l\in[n]} b_{jl} y_{jl} \qquad (8)$$
$$\text{subject to} \quad \sum_{e_l\in V_t} x_l \le k, \qquad \forall t\in[p]$$
$$\sum_{l\in[n]} y_{jl} \le 1, \qquad \forall j\in[m]$$
$$y_{jl} \le x_l, \qquad \forall j\in[m], l\in[n]$$
$$\sum_{e_l\in C_i} x_l \ge \beta\,\mathtt{OPT}_{g_i}, \qquad \forall i\in[d]$$
$$y_{jl} \in \{0,1\}, \qquad \forall j\in[m], l\in[n]$$
$$x_l \in \{0,1\}, \qquad \forall l\in[n]$$

---

[6]https://en.wikipedia.org/wiki/Maximum_coverage_problem
[7]https://en.wikipedia.org/wiki/Optimal_facility_location

Given a set $V = \{e_1, \ldots, e_n\}$ of $n$ items, both ILPs maximize the total *benefit* (i.e., the utility function $f$ in SMFR) provided by a set $S \subseteq V$ for a subset $T \subseteq V$ of $m$ items, subject to representativeness constraints on each $C_1, \ldots, C_d$ subset of $V$ (i.e., the representativeness functions $g_1, \ldots, g_d$ in SMFR). In both formulations, $x_l$ indicates whether $e_l \in V$ is included in the solution $S$, and $y_{jl}$ indicates whether $e_j \in T$ takes the benefit from item $e_l \in V$. Problem 7 is specific to the knapsack constraint defined on a budget $k \in \mathbb{Z}^+$ and a cost function $c(\cdot)$. Problem 8 is specific to the partition matroid constraint, where $V$ is divided into $p$ disjoint partitions $V_1, \ldots, V_p$. For the knapsack constraint, the value of $\texttt{OPT}_{g_i}$ for each representativeness function $g_i$ can be easily computed by sorting the items in $C_i$ ascendingly according to their costs and finding the maximum number of items whose cumulative cost does not exceed $k$. For the partition matroid constraint, the value of $\texttt{OPT}_{g_i}$ for each representativeness function $g_i$ is trivially the maximum between $k$ and $|C_i|$.

## C   Additional Experiments

In this section, we complement the experimental analysis described in Sections 5.1 and 5.2.

### C.1   Additional Experiments on Maximum Coverage

In this section, we use the same data sets and settings as in Section 5.1 for the *Maximum Coverage* problem. For the *Facebook* data set, we alternatively define the knapsack constraint in the same way as for the *DBLP* data set. For the *DBLP* data set, we alternatively define a partition matroid constraint based on the geographic area of the researchers, with five groups: *Asia*, *Europe*, *North America*, *Oceania*, and *South America*. We also set the same upper bound $k \in \mathbb{Z}^+$ for each geographic group, resulting in a partition matroid of rank $r = 5k$. Figures 5 and 6 present the performance of each algorithm for different instances of SMFR on *Maximum Coverage* with knapsack and matroid constraints on the *Facebook* and *DBLP* data sets, respectively. Generally, we observe trends similar to those already presented in Section 5.1, which further confirm the effectiveness of SMFR-SATURATE.

### C.2   Additional Experiments on Recommendation

In this section, we use the same data sets and settings as in Section 5.2 for the *Recommendation* problem. For the *MovieLens* data set, we alternatively define a knapsack constraint by assigning to each item (movie) a random integer cost in the range $[1, 10]$. For the *X-Wines* data set, we alternatively define a partition matroid constraint based on the continent of origin for wine production: *Africa*, *Asia*, *Europe*, *North America*, *South America*, and *Oceania*. We also set the same upper bound $k \in \mathbb{Z}^+$ for each geographic group, resulting in a partition matroid of rank $r = 6k$. Figures 7 and 8 present the performance of each algorithm for different instances of SMFR on *Recommendation* with knapsack and matroid constraints on the *MovieLens* and *X-Wines* data sets, respectively. Generally, we observe trends similar to those already presented in Section 5.2, which further confirm the effectiveness of SMFR-SATURATE.

### C.3   Time Efficiency

Figure 9 reports the running time (in seconds) of SMFR-SATURATE, SATURATE, BSM-SATURATE, and SMSC for SMFR on both *Maximum Coverage* and *Recommendation* instances. We use the same settings as in Sections 5.1 and 5.2. In each plot, the x-axis represents the value of $k$ in the knapsack or matroid constraint and the y-axis represents the running time (in seconds) used by each algorithm to solve an SMFR instance. We present the results for $d = 1$ and $4$ in Figure 9.

All algorithms take less than a minute to complete on each tested instance. SMFR-SATURATE is faster than SMSC in all cases. For the knapsack constraints, SMFR-SATURATE generally runs faster than or close to BSM-SATURATE. However, for the matroid constraints, SMFR-SATURATE is slower than BSM-SATURATE. SATURATE is the fastest method in most configurations. This is because SATURATE does not allow for any trade-off between utility ($f$) and representativeness ($g$) by design and thus is run only once for each instance. However, all other algorithms should be run multiple times with different values of $\beta$ or $\tau$.

### C.4 Experiments on Large Data Sets

**Setup.** To show the applicability of SMFR-SATURATE to data sets larger than those used in Section 5, we performed additional experiments on two larger real-world data sets: *Pokec (Kosicky)* for *Maximum Coverage* and *MovieLens-25M* for *Recommendation.*

The *Pokec (Kosicky)* data set is an undirected graph with $234,320$ nodes and $2,417,175$ edges, extracted from the Pokec[8] data set. Pokec itself is a directed graph that represents the follower-followee relationships of users in a Slovakian social network. *Pokec (Kosicky)* is a subgraph of the Pokec graph induced by nodes representing Pokec users who reside in the Kosicky region while ignoring the directions of the edges. The *MovieLens-25M*[9] data set is a larger version of the *MovieLens* data set presented in Section 5. The only difference relies on the size: the *MovieLens-25M* data set consists of $25,000,095$ ratings from $162,541$ users on $62,423$ movies.

We evaluate the SMFR-SATURATE algorithm and baseline methods on the *Pokec (Kosicky)* data set for the *Maximum Coverage* problem under a knapsack constraint, and on the *MovieLens-25M* data set for the *Recommendation* problem under a partition matroid constraint. We apply the same pre-processing and settings used for the *Maximum Coverage* problem on the *DBLP* data set (Section 5.1) and the *Recommendation* problem on the *MovieLens* data set (Section 5.2) to the *Pokec (Kosicky)* and *MovieLens-25M* data sets, respectively.

**Results.** The top row of Figure 10 presents the performance of each algorithm for different instances of SMFR on *Maximum Coverage* with knapsack constraints on the *Pokec (Kosicky)* data set (see Figures 10a, 10b, and 10c); while the bottom row of Figure 10 presents the performance of each algorithm for different instances of SMFR on *Recommendation* with matroid constraints on the *MovieLens-25M* data set (Figures 10d, 10e, and 10f). Generally, we observe trends similar to those already presented for the other data sets for the same problems under the same kind of constraints.

Figure 11 reports the running time (in seconds) of SMFR-SATURATE, SATURATE, BSM-SATURATE, and SMSC for SMFR on both *Maximum Coverage* instances with knapsack constraints on the *Pokec (Kosicky)* data set and *Recommendation* instances with matroid constraints on the *MovieLens-25M* data set. We use the same settings as in Appendix C.3: In each plot, the x-axis represents the value of $k$ in the knapsack or matroid constraint, and the y-axis represents the running time (in seconds) used by each algorithm to solve an SMFR instance. We present the results for $d = 1$ and $4$ in Figure 11. Generally, we observe trends similar to those in Figures 9b, 9f, 9c, and 9g (see Appendix C.3). Execution times generally grow linearly with the size of the data sets in both applications. The above results have confirmed the applicability of SMFR-SATURATE to larger data sets.

---

[8]https://snap.stanford.edu/data/soc-pokec.html
[9]https://grouplens.org/datasets/movielens/

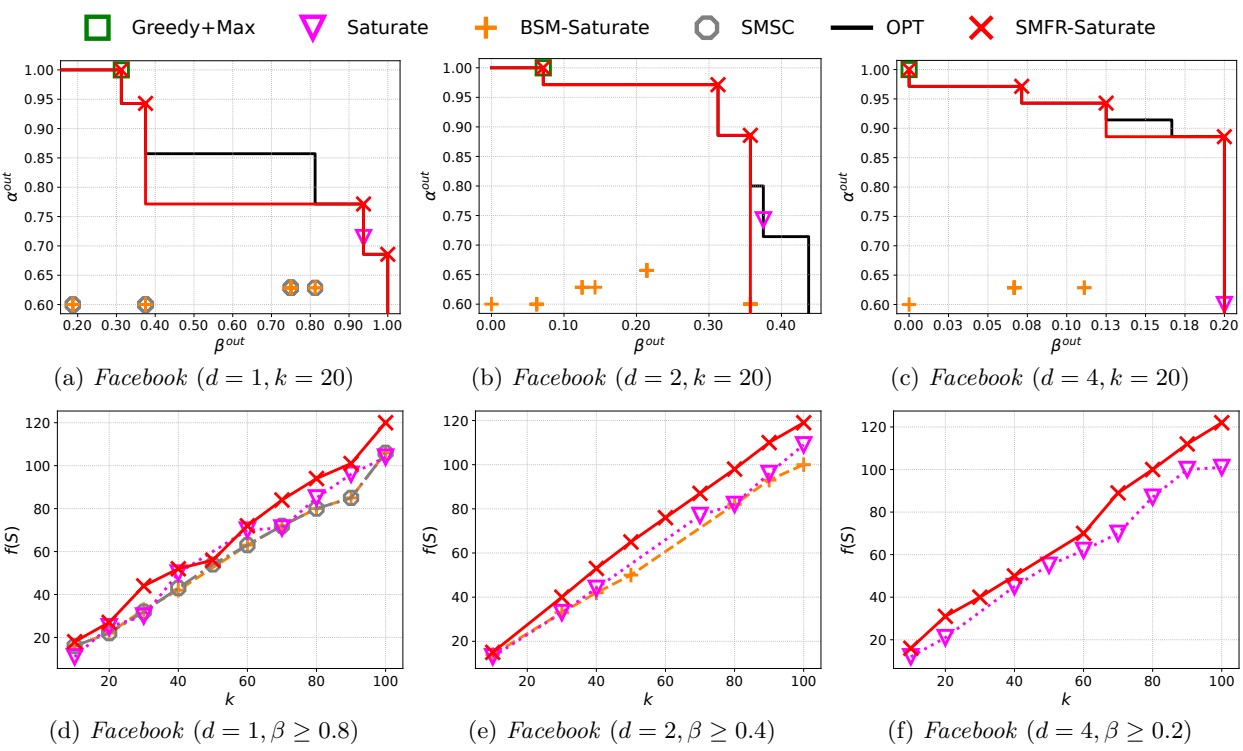

Figure 5: Results for *Maximum Coverage* on the *Facebook* data set, with knapsack constraints.

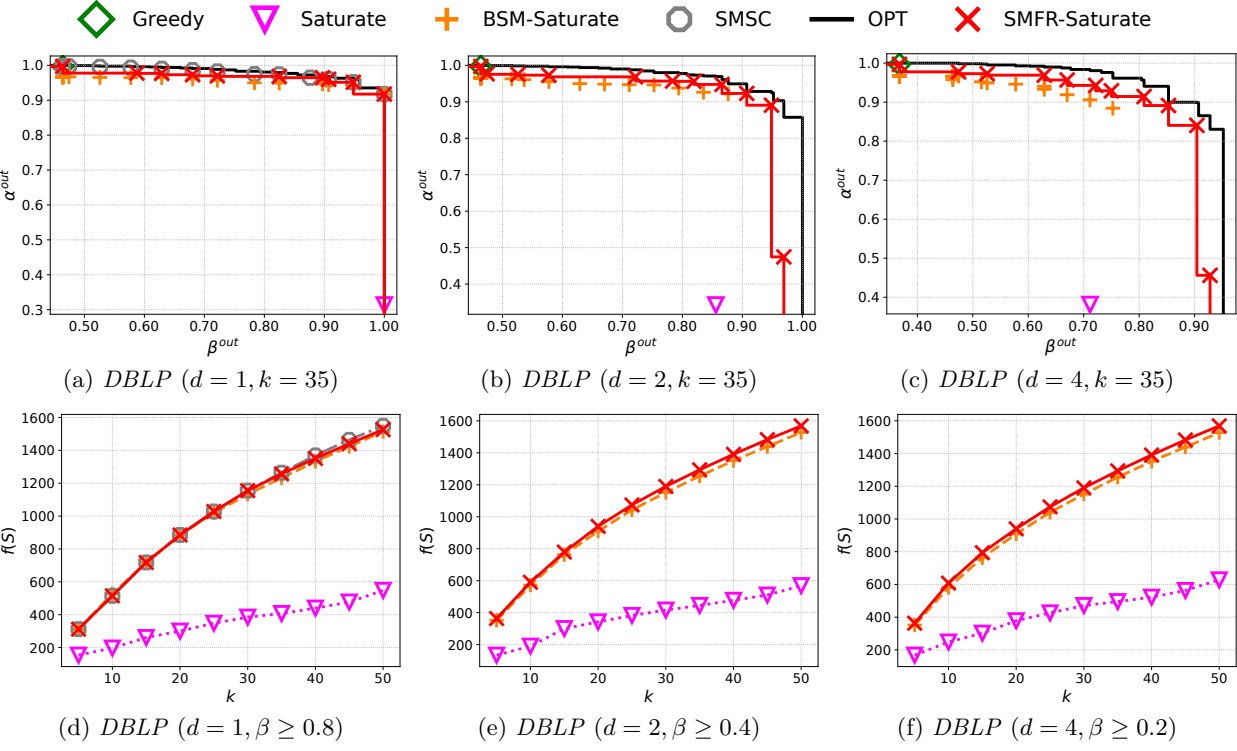

Figure 6: Results for *Maximum Coverage* on the *DBLP* data set, with matroid constraints.

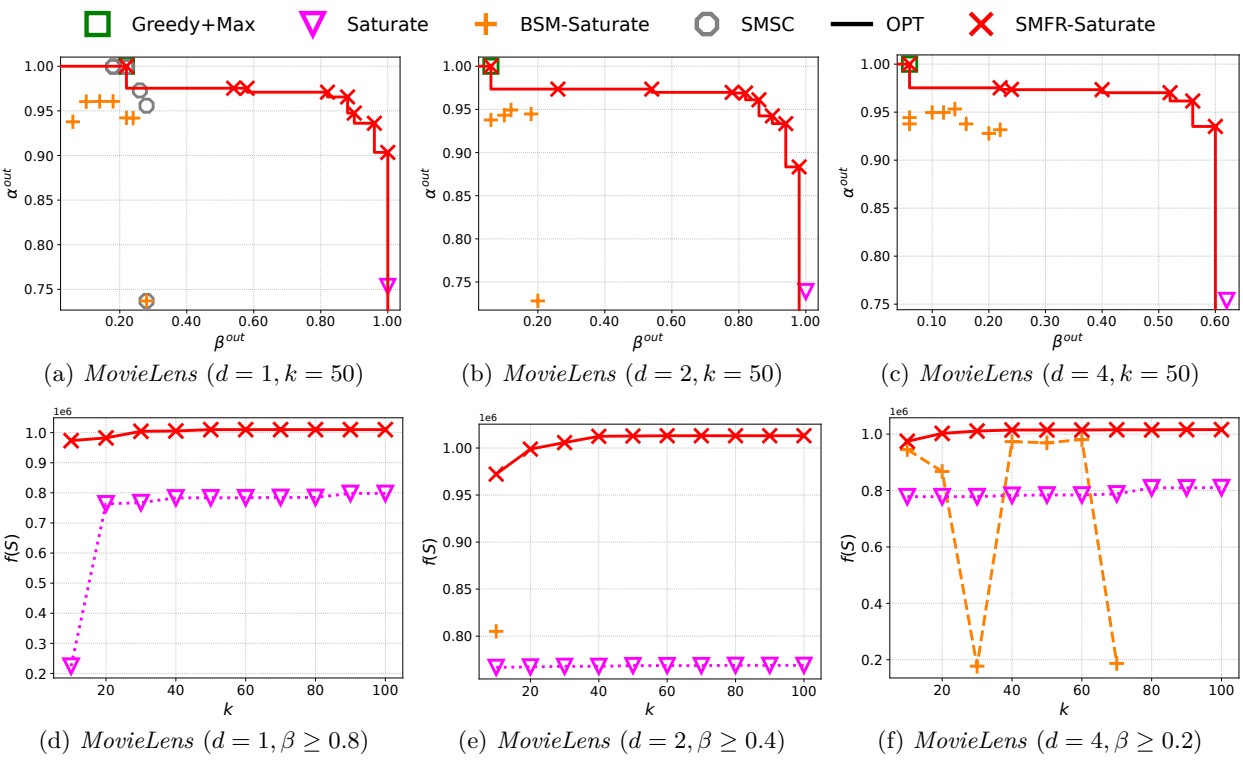

Figure 7: Results for *Recommendation* on the *MovieLens* data set, with knapsack constraints.

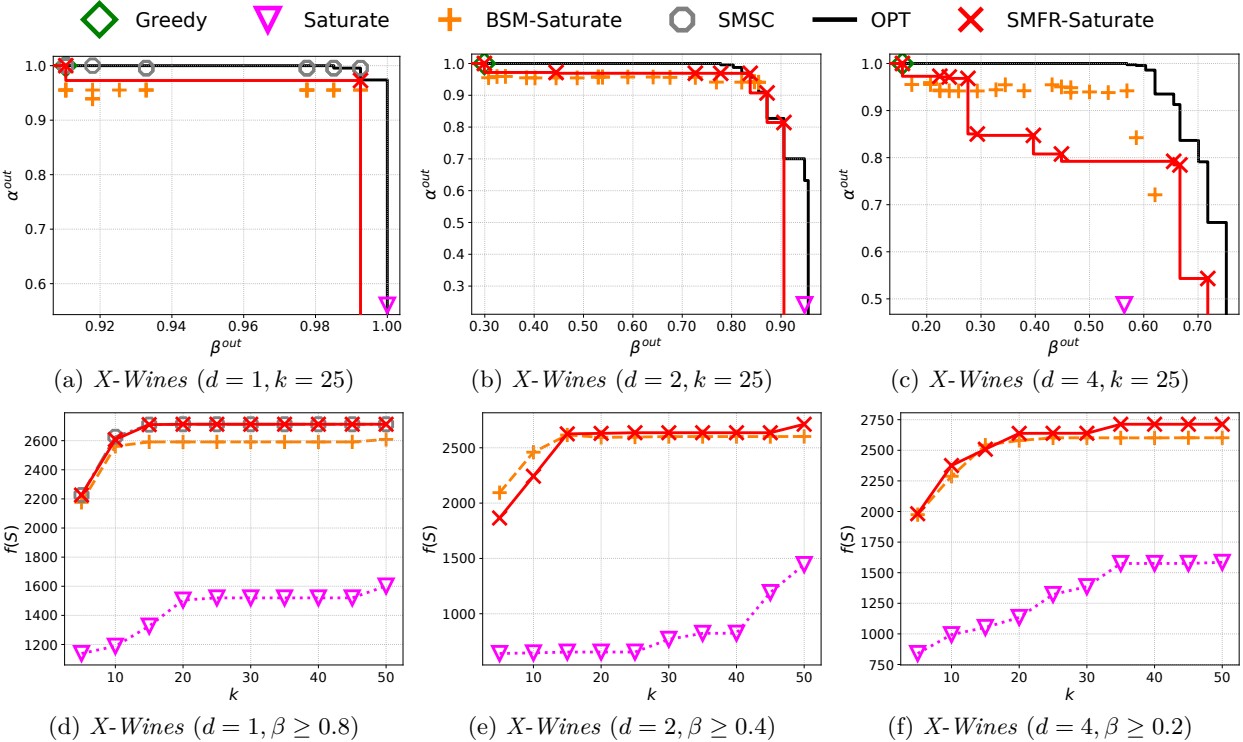

Figure 8: Results for *Recommendation* on the *X-Wines* data set, with matroid constraints.

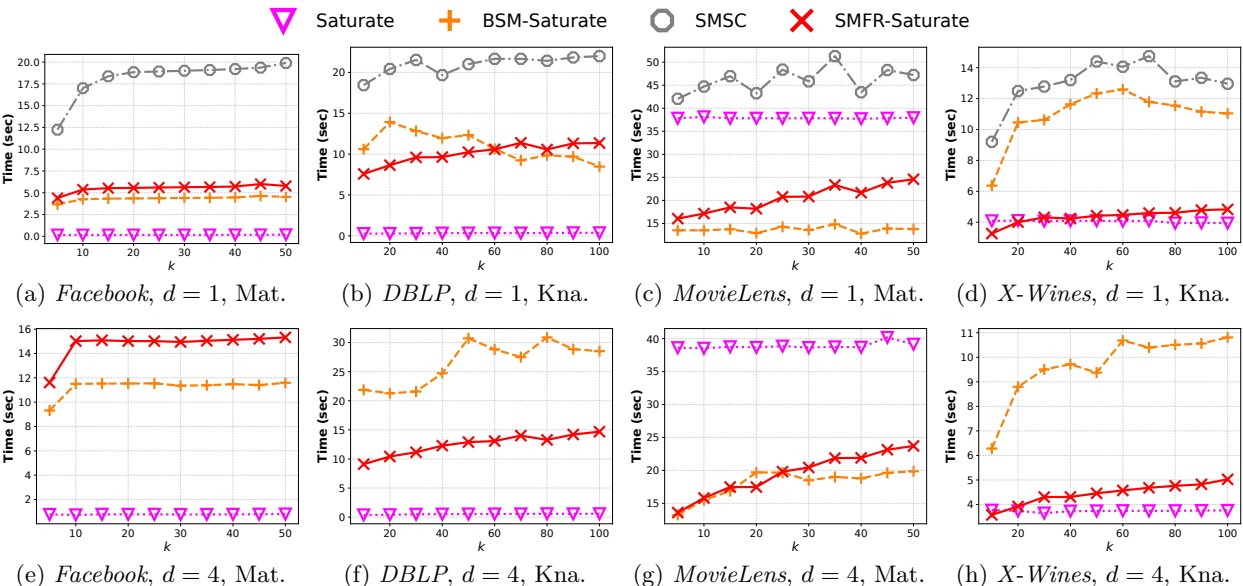

Figure 9: Running times (in seconds) of SMFR-SATURATE, SATURATE, BSM-SATURATE, and SMSC for SMFR when $d = 1, 4$. Here, the *Facebook* and *DBLP* data sets are used for *Maximum Coverage* (MC); the *X-Wines* and *MovieLens* data sets are used for *Recommendation* (RE). In addition, the matroid constraints (Mat.) are imposed on the *Facebook* and *MovieLens* data sets; the knapsack constraints (Kna.) are imposed on the *DBLP* and *X-Wines* data sets.

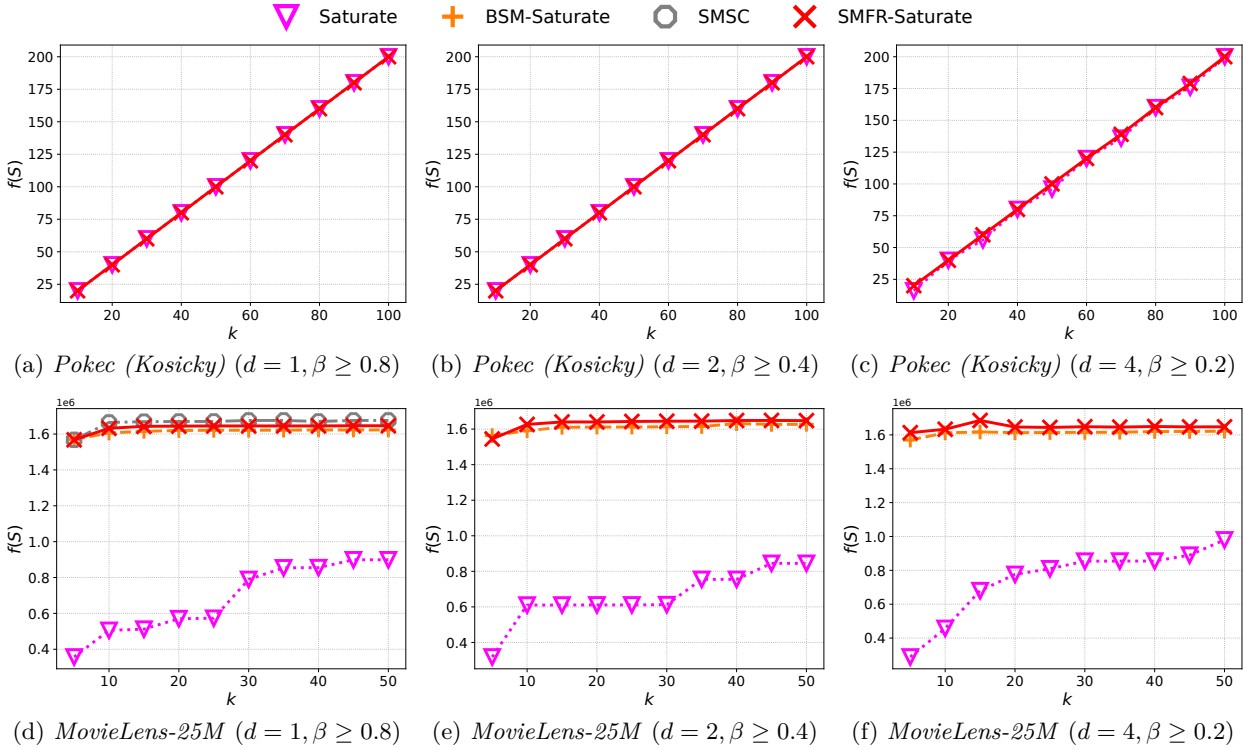

Figure 10: Results for *Maximum Coverage* on the *Pokec (Kosicky)* data set with knapsack constraints, and for *Recommendation* on the *MovieLens-25M* data set with matroid constraints.

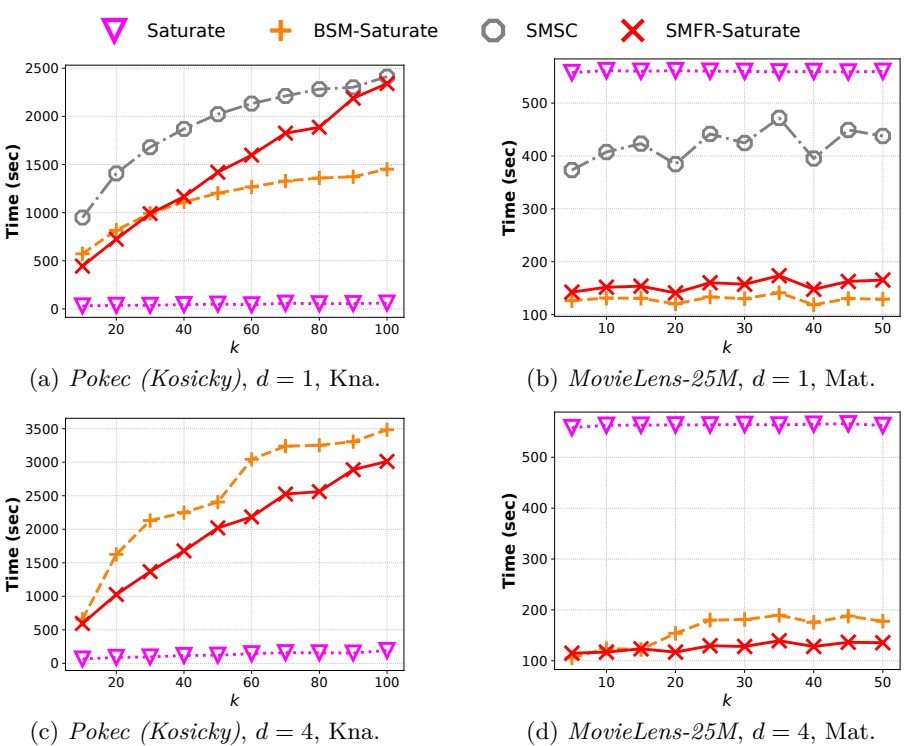

(a) *Pokec (Kosicky)*, $d = 1$, Kna.

(b) *MovieLens-25M*, $d = 1$, Mat.

(c) *Pokec (Kosicky)*, $d = 4$, Kna.

(d) *MovieLens-25M*, $d = 4$, Mat.

Figure 11: Running times (in seconds) of SMFR-SATURATE, SATURATE, BSM-SATURATE, and SMSC for SMFR when $d = 1, 4$. Here, the *Pokec (Kosicky)* data set is used for *Maximum Coverage* (MC) with knapsack constraints (Kna.); the *MovieLens-25M* data set is used for *Recommendation* (RE) with matroid constraints (Mat.).

