# OpenReview forum: "Fair Representation in Submodular Subset Selection: A Pareto Optimization Approach"
_TMLR — Accepted by TMLR_

### Review · Reviewer_6VLx · 2024-02-20

**Summary Of Contributions:**

The paper solves the problem of submodular function maximization under a knapsack or matroid constraint. Unlike previous work, the paper is not maximizing a single objective function but rather a main function f in addition to a collection of d representation functions g_1,..., g_d. The authors suggest finding an (alpha,beta) Pareto optimal solution where the alpha and beta are the approximation factors for f and g1,..,g_d respectively. The main technical result is that an approximation algorithm for submodular maximization under knapsack or matroid constraints can be used to obtain a Pareto optimal approximate solution with close approximation factors. The paper further shows experimental results on several datasets and shows that the proposed algorithm has better performance in comparison to some baselines.

**Audience:**

Yes

**Broader Impact Concerns:**

I don't think that a statement on ethical implications is needed.

**Claims And Evidence:**

Yes

**Requested Changes:**

Please see the points above under weaknesses. The paper would be much stronger if it could convince the reader that other more traditional options such as maximizing a weighted sum or imposing bounds are not the right thing to do here.

**Strengths And Weaknesses:**

### Strengths:
-I think the topic is important and interesting and it is nice to see that previous work can be used to obtain an approximate solution to this setting.

-The experiments have been broad comparing to various baselines and datasets.

### Weaknesses:
W1-The major weakness in my view has to do with the problem formulation. Specifically, Pareto optimality is in general a very weak guarantee. Why haven’t the authors considered other possibilities such as imposing bounds on the functions g_1, …, g_d or maximizing a weighted sum of the functions? These possibilities and their disadvantages should be discussed in the paper.

W2-The paper would be more clear if the point behind functions g_1, …, g_d is clarified early on and separated from the knapsack/matroid constraints. Example 1 in the paper makes it seem as though the constraints (knapsack or matroid) are there to ensure fairness, but further ahead in the paper this is done by the representativeness objectives  g_1, …, g_d

W3-In the related work, the paper states that the previous work “returns only a single solution”. But is this not the case in this paper as well?


W4-This issue is also about the Pareto optimality, consider the knapsack constraint in section 5.1. The functions g_i are counting vertices from different communities, can’t we end up with trivial solutions where for example one group has full representation and the other has none? Would that not be Pareto-optimal? What forbids this? This would not happen on the other hand if we imposed bounds.

---

> ### Author Response · Authors · 2024-03-06
> **Response from the Authors**
>
> Thanks for your valuable comments. Next, we will respond to all the points that you raised and indicate the revisions we have made to address them.
>
> ### **[Strenghts S1-S2]**
>
> We sincerely thank the Reviewer for recognizing the interestingness of our work and the quality of our experimental assessment.
>
> ### **[Weakness - W1]**
>
> First, although finding _a single_ Pareto-optimal solution might be a weak guarantee for multi-objective optimization problems (for example, a solution that maximizes one objective function while ignoring all the others is Pareto optimal), our goal is to find the approximations for \emph{all} Pareto-optimal solutions and thus to approximate the Pareto frontier, which is a non-trivial and much stronger goal.
>
> In our approach, the value of $\beta$ indicates the bound imposed on $g_1, \dots, g_d$ since an $(\alpha, \beta)$-approximate solution ensures at least a $\beta$-approximation w.r.t. each of $g_i$ for every $i \in [d]$. As shown in the related work, the formulations of SMSC and BSM also impose a bound on $g_1, \dots, g_d$. But different from SMFR, they pre-specify a given bound on $g_1, \dots, g_d$ as a constraint on maximizing $f$ and do not exploit how to trade off among them. Alternatively, SMFR attempts to find different values of $\alpha$ and $\beta$ and the corresponding solutions to approximate all Pareto-optimal ones. Therefore, SMFR significantly generalizes existing multi-objective submodular maximization problems such as SMSC and BSM and imposes more flexible bounds on $g_1, \dots, g_d$.
>
> Maximizing a weighted sum of objective functions is the first and simplest formulation for multi-objective submodular maximization problems. Since the non-negative linear combination of submodular functions is still a submodular function, it can be directly solved by any algorithm for submodular maximization. However, maximizing a weighted sum of objective functions is not a meaningful formulation when we want to achieve a fair representation of $g_1, \dots, g_d$ because the solution that maximizes the weighted sum may not have any approximation guarantee for each function individually. In addition, it is also challenging to decide the weights of different objective functions to meet various needs. Contrarily, SMFR does not require any input to decide the weights of objective functions, and its solutions can provide approximation guarantees for all objective functions.
>
> Thanks for this fruitful discussion: in the revised paper, we discuss the disadvantages of these choices compared to SMFR.
>
> ### **[Weakness - W2]**
>
> We note that the knapsack/matroid constraints are not used to achieve fair representation in our setting but are to impose a restriction on a single solution, as for general submodular maximization problems. We have revised Example 1 to clarify this point.
>
> ### **[Weakness - W3]**
>
> This is not the case for SMFR. The goal of SMFR is to provide a collection of (multiple) solutions with different trade-offs between $f$ and $g_1, \dots, g_d$ so that they can approximate all Pareto-optimal solutions.
>
> ### **[Weakness - W4]**
>
> There may exist a Pareto-optimal solution that maximizes $f$ but does not provide any approximation guarantee on any $g_i$, which corresponds to a $(1, 0)$-approximate Pareto-optimal solution according to our definition. The $(1, 0)$-approximate solution can be included in the collection of solutions (i.e., an extreme case that prioritizes $f$ but ignores $g_1, \dots, g_d$) returned by our algorithm, but we do not need to avoid this case. This is because our algorithm will also try to find other solutions with different approximation factors $\alpha$ and $\beta$, which correspond to different trade-offs between $f$ and $g_1, \dots, g_d$.
>
> ### **[Request - R1]**
>
> We have carefully responded to the above points and revised the paper according to them.

---

### Review · Reviewer_bDDZ · 2024-02-23

**Summary Of Contributions:**

The paper suggests the so-called “submodular maximization with fair representation (SMFR)” problem and obtains Pareto approximation algorithms for the problem subject to knapsack and matroid constraints. In this problem, one is given a main (monotone) submodular function f, and d other submodular functions g_1, …, g_d. A subset S is called (alpha, beta)-approx, if it is alpha-approx to the max of main function f, and beta-approx to the max of all the other d functions g_1, …, g_d simultaneously. Roughly speaking, the Pareto approximation is a set P of solutions, such that for every alpha, beta, if there is a (alpha, beta)-approx solution S’ then there is a (\delta alpha - eps, \delta beta - eps)-approx to it in the solution set P, where \delta is the approx-ratio of a black-box of an algorithm for submodular maximization subject to the matroid or knapsack constraints. The main result is that |P| = O(1 / eps) is sufficient for this goal, and the every solution set in P is of small size (linear in the size of the knapsack or the rank of the matroid). This can be found efficiently in poly time. Experiments are conducted to validate the effectiveness of the algorithm on real datasets.

**Audience:**

Yes

**Broader Impact Concerns:**

None.

**Claims And Evidence:**

Yes

**Requested Changes:**

- \mathcal{I} is not quantified in (1). Maybe you want to change to “subject to a knapsack or matroid constraint \mathcal{I}”, in the sentence immediately before (1)?

- The meaning of max in (1) is not discussed in the paragraph immediately below it. While I understand that the exact definition is discussed in ”Our Contributions” part, it is still suggested to at least comment that the intuition is to maximize them simultaneously even though this goal can be tricky and you would discuss this next.

- Page 2, I think “and at least one is strictly larger” should be put out of the parenthesis since this is an important part of the definition. And similarly for the one on page 5.

- In the statement of lemma 1, the quantification of S sounds confusing. I suppose you meant “exists S \in \mathcal{I}”, instead of “for every S”. However, currently it reads “for any set S \in \mathcal{I}”, and it may confuse with “for every S”. Maybe replace “any” with “some”? Or, rephrase as e.g. “suppose S \in \mathcal{I} is any set that satisfies F’_{\alpha, \beta}(S) \geq d + 1 - eps / 2”

**Strengths And Weaknesses:**

Strength:

- The writing quality is good

- The result of only O(1 / eps) solutions can already capture the Pareto frontier looks interesting and strong

- The citations and comparison to related works are comprehensive

- The experiments are solid


Weakness:

- The technical novelty is limited, and it seems to be a composition of several existing results in a relatively straightforward way

- The size of the data sets used in the experiments looks small

---

> ### Author Response · Authors · 2024-03-06
> **Response from the Authors**
>
> Thanks for your valuable comments. Next, we will respond to all the points that you raised and indicate the revisions we have made to address them.
>
> ### **[Strenghts S1-S4]**
>
> We sincerely thank the Reviewer for recognizing the interestingness and solidity of our results, the quality of the presentation and the experimental assessment.
>
> ### **[Weakness - W1]**
>
> We note that another major technical novelty of our work is to propose a new Pareto optimization approach to multi-objective submodular maximization (i.e., SMFR), which differs from the existing approaches to multi-objective submodular maximization and existing Pareto optimization approaches to (single-objective) submodular maximization, as we analyze in the Introduction and Related Work sections. Due to the hardness of SMFR and as the first attempt along this new direction, we study how to transform it into a known problem (i.e., submodular cover) and extend existing algorithms for submodular cover to solve the SMFR problem. Although the idea of the algorithm design is simple, the theoretical analysis for it is non-trivial because SMFR has distinct properties from the original submodular cover problem, so directly applying the existing algorithms cannot work for SMFR. Finally, we acknowledge that our algorithms and theoretical results are far from being the ultimate solution: further investigation into algorithmic techniques for SMFR, along with tighter theoretical results, are needed and will be part of our future endeavors.
>
> ### **[Weakness - W2]**
>
> We used small data sets in the original submitted manuscript because the ILP solver cannot provide optimal solutions in a reasonable time for large instances. However, our algorithm can certainly scale to much larger data sets. In the revised paper, we have included additional experimental results on two larger data sets (Pokec (Kosicky) for _Maximum Coverage_ and MovieLens-25M for _Recommendation_) in Appendix C.4. The additional results show that our algorithm can still provide high-quality results on the two large data sets in less than one hour.
>
> ### **[Request - R1]**
>
> Thank you. We have fixed this point by explicitly indicating the meaning of $\mathcal{I}$ before using it in Eq. (1).
>
> ### **[Request - R2]**
>
> Thank you. We have fixed this point by explaining what “max'' in Eq. (1) means and why several objective functions should be maximized simultaneously in the paragraph immediately below Eq. (1) in the revised paper.
>
> ### **[Request - R3]**
>
> We have changed the expression following your helpful suggestion.
>
> ### **[Request - R4]**
>
> We are very grateful to you for pointing out the issue: we now realize that our expression in Lemma 1 may be confusing. We have revised Lemma 1 accordingly.

---

### Review · Reviewer_vmk1 · 2024-02-25

**Summary Of Contributions:**

This work introduces the problem of submodular maximization with fair representation (SMFR).
In particular, it aims to maximize a (normalized) monotone submodular function
subject to a knapsack or matroid constraint in a Pareto-optimal way with
respect to $d$ representative objective functions $g_i$.
The main idea is to recast an SMFR instance into mutiple instances of the
submodular cover problem with different weights between the $f$ and $g_i$
objectives.

The authors show that if we have a $\delta$-approximation algorithm for
submodular maximization with a knapsack (or matroid) constraint that returns a
$(\alpha,\beta)$-approximate Pareto optimal solution to SMFR, then there exists
a $(\delta \alpha - \varepsilon, \delta \beta - \varepsilon)$-approximate
solution with cost $O(k \log (d \ \varepsilon))$ where $k$ is the knapsack
budget (or matroid rank) constraint. They also provide extensive experiments on
real-world data that compares their algorithm on max coverage and
recommendation tasks to true optimal solutions computed via integer
programming.

**Audience:**

Yes

**Claims And Evidence:**

Yes

**Requested Changes:**

**Questions**
- [page 02] In Eq. (1), how is the max operator defined for a tuple of real
  values? If we use a lexicographic comparator, then the first representative
  function $g_1$ carries more weight than $g_d$.
- [page 02] In the definition of Pareto optimal for $(\alpha,
  \beta)$-approximate solutions, instead of saying "there does not exist any
  ...", should you instead relax this to "you have not computed any" (i.e., a
  solution is Pareto-optimal w.r.t. a set of computed solutions? Otherwise, this
  quantifier includes optimal solution that you may not have computed. The same
  applies to the top of page 05.

**Typos / suggestions**
- [page 02] Consider putting Example 1 in a \paragraph so that the text is not
  all italcized, as it is somewhat hard to read for more than a few lines.
- [page 04] Typo: "which ensures that the difference in the utilities of any two groups is As ..."
- [page 04] In the beginning of Section 4, you do not need to reintroduce what
  SMFR means, as it was defined earlier.

**Strengths And Weaknesses:**

**Strengths**
- This work proposes an interesting a novel way to maximize monotone submodular
  functions in the presence of structured constraints and $d > 1$
  "representative" functions. In particular, this differs from past works that
  simply aim to maximize the *minimum* of the "fairness" objectives $g_i$.
- The experiments are quite strong for an applied paper that studies submodular
  maximization.

**Weaknesses**
- The only concrete theoretical contribution is Theorem 1 and Theorem 2, which
  formalize how to recast the SMFR to a sequence of submodular cover problems.

---

> ### Author Response · Authors · 2024-03-06
> **Response from the Authors**
>
> Thanks for your valuable comments. Next, we will respond to all the points that you raised and indicate the revisions we have made to address them.
>
> ### **[Strenghts]**
>
> We sincerely thank the Reviewer for recognizing the interestingness and novelty of our proposal and the quality of the experimental assessment.
>
> ### **[Weakness - W1]**
>
> We agree with the Reviewer that Theorems 1 and 2 are the main theoretical results of our algorithm for SMFR in the case of knapsack and matroid constraints, respectively. However, we note that those theoretical results are not trivial considering the distinct properties of SMFR from existing problems. For the sake of the paper’s readability, we decided to defer most of the analysis to Appendix A. Moreover, we also stress the importance of Lemmas 1 and 2 which confirm the correctness of transforming any approximation algorithm for the submodular cover problems w.r.t. SMK or SMM into an approximation for SFMR.
>
> ### **[Question - Q1]**
>
> Eq. (1) follows the convention to formulate multi-objective optimization problems, which means that all objective functions, i.e., $f(\cdot), g_1(\cdot), \dots, g_d(\cdot)$ should be maximized _simultaneously_ without any specified order. Since the goal of Eq. (1) is shown to be infeasible, we alternately find Pareto-optimal solutions. However, we do not impose a lexicographic order on the objective functions or use a lexicographic method to find Pareto-optimal solutions. In our approach, the objective functions are divided into two groups: (1) the utility function $f$ and (2) the representativeness functions $g_1, \dots, g_d$. Pareto optimality is defined w.r.t. the trade-off between maximizing $f$ (as quantified by $\alpha$) and maximizing all $g_i$'s (as quantified by $\beta$). Therefore, in our formulation and algorithms, $g_1(\cdot), \dots, g_d(\cdot)$ are always treated equally because each of them has the same effect on the value of $\beta$.
>
> ### **[Question - Q2]**
>
> We confirm that our definition of _Pareto optimality_ is valid and consistent with similar concepts in other multi-objective optimization problems, and the relaxation is not necessary. In fact, it is feasible to guarantee that the (approximate) Pareto-optimal solutions w.r.t.~all possible solutions without explicitly computing any of them. This is because our theoretical analysis for (approximate) Pareto optimality does not rely on comparing the computed solutions with any other solutions. It relies on the approximation bounds of the greedy algorithms and by contradiction: If there is a solution that makes the computed solution violate its approximate Pareto optimality, then the approximation bound of the greedy algorithms cannot hold, leading to a contradiction.
>
> ### **[Typos / suggestions]**
>
> Thanks a lot for all these recommendations that we have implemented in the revised version.

---

### Author Response · Authors · 2024-03-06
**Revised Version Uploaded**

Dear AE and Reviewers,

Thank you for the time spent in managing and reviewing our work. We have already uploaded a revised version of our paper aimed at clarifying the motivations and contributions of our work, as well as other comments raised by the reviewers.

To facilitate the review of the modifications, new content in the manuscript is highlighted in blue. We next proceed to reply to the reviews, by means of a separate comment for each review.

Sincerely,
The Authors

---

### Author Response · Authors · 2024-03-15
**Further points for discussion?**

Dear AE, dear Reviewers,

We sincerely thank you for your effort in managing and reviewing our submission. We would like to ask the Reviewers, whether they are satisfied with our responses or if there is any further point to discuss in this phase.

Many thanks, The Authors.

---

### Decision · Action_Editor_4n4P · 2024-05-16

**Recommendation:** Accept with minor revision

**Comment:**

The reviewers' consensus was that the proposed formulation is likely to be of limited interest. However, the formulation still makes sense, and the results are accurate and correct. So I think regarding the "accuracy, credibility and clarity", the paper is still doing good enough. They also agree that regardless of the technical depth and potential impact, the second criteria (for audience interest) is met.

As a result, I would like to ask the authors to include a more thorough discussions of the advantages and limitations of this approach, as done through the rebuttal.

**Audience:**

The reviewers agree that the audience is limited, but non-empty.

**Claims And Evidence:**

The reviewers found the paper and its technical claims to be accurate and well-written. However, they also found the motivation and evidence, both analytical and empirical, in favor of utilizing the Pareto approach to be currently insufficient. As a result, I would like to ask the authors to include a more thorough discussions of the advantages and limitations of this approach, as done through the rebuttal.